# Online searches to evaluate misinformation can increase its perceived veracity

Kevin Aslett[1 ✉], Zeve Sanderson[2], William Godel[2], Nathaniel Persily[3], Jonathan Nagler[2,4] & Joshua A. Tucker[2,4]

Considerable scholarly attention has been paid to understanding belief in online misinformation[1,2], with a particular focus on social networks. However, the dominant role of search engines in the information environment remains underexplored, even though the use of online search to evaluate the veracity of information is a central component of media literacy interventions[3–5]. Although conventional wisdom suggests that searching online when evaluating misinformation would reduce belief in it, there is little empirical evidence to evaluate this claim. Here, across five experiments, we present consistent evidence that online search to evaluate the truthfulness of false news articles actually increases the probability of believing them. To shed light on this relationship, we combine survey data with digital trace data collected using a custom browser extension. We find that the search effect is concentrated among individuals for whom search engines return lower-quality information. Our results indicate that those who search online to evaluate misinformation risk falling into data voids, or informational spaces in which there is corroborating evidence from low-quality sources. We also find consistent evidence that searching online to evaluate news increases belief in true news from low-quality sources, but inconsistent evidence that it increases belief in true news from mainstream sources. Our findings highlight the need for media literacy programmes to ground their recommendations in empirically tested strategies and for search engines to invest in solutions to the challenges identified here.

Concern over the impact of misinformation has continued to grow, as high levels of belief in misinformation have threatened democratic legitimacy in the United States[1] and global public health during the COVID-19 pandemic[2]. Considerable attention among scholars, media and policymakers alike has been paid to the role of social media platforms in the spread of, and belief in, misinformation[3,4], with comparatively little focus on other central features of the digital information ecosystem.

This gap in research is particularly evident in our limited understanding of the effect of search engines. Although recent research has explored the potential partisan biases of search engine results[5–7], relatively little is known about the fundamental but understudied question of how searching online to evaluate news (SOTEN) impacts belief in misinformation. As the cost of producing and distributing information online has fallen and the sheer volume of information on the internet has risen, reliance on traditional gatekeepers has been substantially reduced, leaving search engines to fill the role of twenty-first-century gatekeepers by sorting and validating online content for the public[8,9]. In this new role, search engines have become influential in users' political knowledge[10] and public opinion[9]. A majority of internet users state that they check facts online that they come across at least once a day, and many believe that results from search engines are more reliable than traditional news, such as radio, newspapers or television[11].

The growing reliance on search engines for information verification has been encouraged by social media companies[12], civil society organizations[13] and government agencies[14], all of which have invested in campaigns to encourage online users to research news they believe may be suspect through online search engines with the goal of reducing belief in misinformation. Although search engines have a key role in how people evaluate information online, we know little about how SOTEN impacts belief in misinformation.

Research on interventions designed to mitigate belief in misinformation has developed in recent years, but work has thus far focused on ideological congruence[15,16], psychological factors[17,18] and digital media literacy[19]. Here we present the results from experimental studies identifying how SOTEN affects belief in misinformation. Specifically, we test a preregistered hypothesis that searching online to assess the veracity of false or misleading articles increases the belief that these stories are true, contradicting what we believe to be the received wisdom underlying many search-based recommendations. We then examine a possible mechanism for why belief in false/misleading articles is increased by searching online to evaluate these articles: exposure to unreliable information. Although it is plausible that searching online may lead respondents to reputable sources contradicting the false article's central claim, previous studies on information systems have suggested that there are topics or terms for which there exists

[1]School of Politics, Security and International Affairs, University of Central Florida, Orlando, FL, USA. [2]Center for Social Media and Politics, New York University, New York, NY, USA. [3]Stanford University Law School, Stanford University, Stanford, CA, USA. [4]Wilf Family Department of Politics, New York University, New York, NY, USA. ✉e-mail: kevin.aslett@ucf.edu

**a**

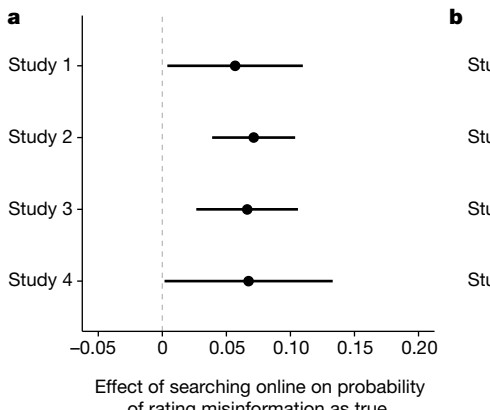

**b**

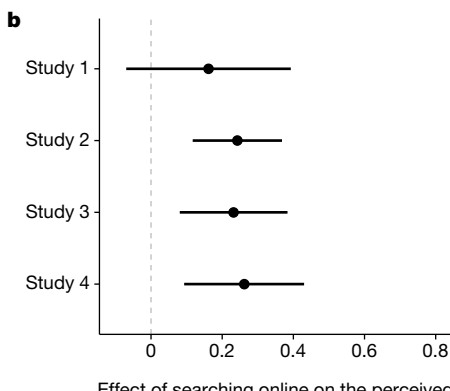

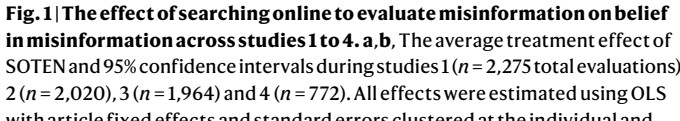

Effect of searching online on probability of rating misinformation as true

Effect of searching online on the perceived veracity of misinformation (7-point scale)

**Fig. 1 | The effect of searching online to evaluate misinformation on belief in misinformation across studies 1 to 4. a,b**, The average treatment effect of SOTEN and 95% confidence intervals during studies 1 ($n$ = 2,275 total evaluations), 2 ($n$ = 2,020), 3 ($n$ = 1,964) and 4 ($n$ = 772). All effects were estimated using OLS with article fixed effects and standard errors clustered at the individual and article level. **a**, The effect of SOTEN on rating misinformation as true for studies 1 ($P$ = 0.037), 2 ($P$ < 0.0001), 3 ($P$ = 0.0018) and 4 ($P$ = 0.0451). **b**, The effect of SOTEN on a seven-point ordinal scale of veracity for studies 1 ($P$ = 0.154), 2 ($P$ = 0.0004), 3 ($P$ = 0.0038) and 4 ($P$ = 0.0054).

unreliable information available to be returned by search engines[20]. As a number of digital literacy guides focus specifically on identifying misinformation, our main analyses are limited to the effect of search on belief in misinformation; however, given that the average online media diet comprises substantially more true than false news[21–23], we also test a preregistered hypothesis that searching online to assess the veracity of true articles increases belief in those articles.

To this end, we run five separate experiments that measure the effect of SOTEN on belief in popular false and true news stories for the point in time investigated. Four of these studies use survey experiments; the fifth combines survey and digital trace data of search results collected using a custom web browser plug-in. In each study, the individuals in both the control and treatment groups were asked to assess the veracity of news articles, but those in the treatment group were encouraged to search online for information (instructions to search online were provided by a partner organization and are provided in the Methods) to help with this assessment. In an additional experiment, explained in Supplementary Information O, we tested whether the effect of SOTEN was robust to changing the wording of these instructions and found similar effects (Extended Data Fig. 1). For all five studies, we used a pipeline (which was also preregistered) to select popular articles from both main-stream and non-mainstream media sources and then distribute them to respondents and professional fact-checkers (a full explanation of this process is provided in the Methods). A key feature of our design is the ability to collect real-time evaluations in the time period during which past research has shown that misinformation is most likely to be consumed[24–26].

Taken together, the five studies provide consistent evidence that SOTEN increased belief in misinformation during the point in time investigated. In our fifth study, which tested explanations for the mechanism underlying this effect, we found evidence suggesting that exposure to lower-quality information in search results is associated with a higher probability of believing misinformation, but exposure to high-quality information is not. Moreover, we found that there is a search effect on belief in true news that is similar to the search effect on belief in false/misleading news: searching can make study participants more likely to believe that true news stories are true. However, when we subset the results by the quality of source, we found that, although online search can increase belief that true news from low-quality sources is true, there is no consistent effect in either direction on believing true news from mainstream sources is true.

## Measuring the online search effect

Our first study (study 1) tests the effect of SOTEN on belief in misinformation using a randomized controlled trial. We recruited 3,006 respondents living in the United States through Qualtrics, an online survey firm, over 10 days and presented the participants with three articles from mainstream and low-quality sources within 48 h of publication (more details about the respondent recruitment and article selection are provided in the Methods). The participants were either randomly assigned to be encouraged to search online to help them to evaluate all of the articles that they were sent (treatment group) or they were not prompted to search online (control group). All of the respondents were then asked to evaluate the veracity of the article using both a categorical (true, false/misleading, could not determine) and seven-point ordinal scale. A key challenge was establishing the veracity of the articles directly after publication, a period during which assessments from fact-checking organization were not likely to be available. To this end, we sent out the articles to be evaluated concurrently by a group of six professional fact-checkers from leading national outlets. The fact-checkers could rate articles as 'true', 'false or misleading' or 'could not determine'. Each article was then labelled as either 'true', 'false or misleading' or 'could not determine' based on the modal fact-checker evaluation. In this section, we analyse only the effect of searching online on belief in articles labelled as false/misleading. During study 1, across 13 false/misleading news articles, we collected 1,145 evaluations from 876 unique respondents in the control group and 1,130 evaluations from 872 unique respondents in the treatment group. Details about all of the articles in each study are provided in Supplementary Tables 1–22 in Supplementary Information A. The number of unique respondents and evaluations in studies 1–5 are provided in Supplementary Tables 76–80.

To estimate the treatment effect of being encouraged to search online, we fit an ordinary least squares (OLS) regression model with article-level fixed effects and standard errors clustered at the respondent and article level to predict belief in misinformation (that is, rating a false or misleading article as true). For our dichotomous outcome, OLS or logistic regressions produce similar results and are both appropriate, but an OLS regression is preferred to estimate the causal effects of treatments on a binary outcome[27]. We control for basic demographic factors (age, education, income, ideological congruence and gender) and, unless noted otherwise, all of the models in this Article follow these specifications. Figure 1a shows that being encouraged to search online increased the probability that a respondent rated a false or misleading article as true by 0.057 ($P$ = 0.037, Cohen's $D$ = 0.12, $n$ = 2,275).

This represents a 19% increase in the probability that a respondent rated a false or misleading article as true. Figure 1b shows a 0.16 increase in perceived veracity using a seven-point ordinal scale ($P = 0.154$, Cohen's $D = 0.09$, $n = 2,275$). Supplementary Tables 23–66 in Supplementary Information B display the full regression results for all of the models.

We next examined whether the search effect was strong enough to change an individual's evaluation after they had already assessed the veracity of a news story. To do so, we ran a within-respondent study (study 2) in which we first asked the respondents to evaluate an article without encouraging them to search online, and then asked the respondents to evaluate the same article again, but after encouraging them to search online. If we assume that the respondents have a bias towards consistency, this offers an even stronger test than in study 1 because, to find a search effect, the respondents would have to change their previous evaluation. To conduct the study, we recruited 4,252 American respondents through Qualtrics over 33 days, 1,010 of whom were presented with one false/misleading popular online article within 48 h of publication. We then compared their evaluation before being encouraged to search online (control) and their evaluation after being encouraged to do so (treatment). Notably, we also found that, in study 2, searching online increased the probability that a respondent rates a false/misleading article as true by 0.071 ($P < 0.0001$, Cohen's $D = 0.15$, $n = 2,020$), which represents a 22% increase in the likelihood that a respondent thinks that a false news story is true, and a 0.24 ($P = 0.0004$, Cohen's $D = 0.13$, $n = 2,020$) increase on a seven-point ordinal scale. We found that, among those who first rated the false/misleading article correctly as false/misleading, 17.6% changed their evaluation to true after being prompted to search online (for comparison, among those who first incorrectly rated the article as true, only 5.8% changed their evaluation to false/misleading after being required to search online). Among those who could not initially determine the veracity of false articles, more individuals incorrectly changed their evaluation to true than to false/misleading after being required to search online. This suggests that searching online to evaluate false/misleading news may falsely raise confidence in its veracity.

While these first two studies present consistent evidence that searching online can increase belief in misinformation directly after its publication, misinformation can, in some instances, go viral weeks or months after publication. In these cases, the online information environment surrounding the false article could be different from the one encountered in the first 48 h. Directly after publication of false articles, search engines may return similar misinformation and little credible information because professional fact-checks often take days or weeks to be published[28]. We therefore might expect that, as time passes after publication, individuals searching online would be exposed to more professional fact-checks and credible information, potentially eliminating or, even more optimistically, changing the direction of the search effect identified in studies 1 and 2.

To test the robustness of the findings from studies 1 and 2 when more time had passed after publication, we ran a third study (study 3) that replicates study 2 with new respondents evaluating the same set of articles but, this time, between 3 and 6 months after the publication of the articles. For study 3, we recruited 4,042 American respondents over 1 month through Qualtrics, 982 of whom evaluated one false/misleading article first without being encouraged to search online and then again after being encouraged to search online. We found that searching online increases the probability that a respondent rates a false/misleading article as true by 0.066 ($P = 0.0018$, Cohen's $D = 0.14$, $n = 1,964$), which means that 18% more respondents rated the same false/misleading story as true after they were asked to re-evaluate the article after treatment, even months after the article was published. We also found that searching online leads to a 0.23 point increase on a seven-point scale ($P = 0.0038$, Cohen's $D = 0.13$, $n = 1,964$). Although it may be possible that respondents were exposed to more reliable information months after publication, it does not appear to have negated the impact of SOTEN on belief in misinformation.

The first three studies measured the effect of SOTEN on popular pieces of misinformation, which may cover niche topics that are not reported on by reliable news outlets. However, we might expect a different and, hypothetically, more reliable news environment when searching online about more salient events, such as the COVID-19 pandemic.

On the one hand, substantial reporting from reliable sources on this topic are more likely to be available, which could reduce the effect of SOTEN on belief in misinformation. On the other hand, it is possible that highly salient events also attract more misinformation, for either political or economic reasons[29]. Thus, to determine whether the effect of SOTEN on belief in misinformation holds when researching misinformation about a salient event, we ran a fourth study (study 4) during the heart of the COVID-19 pandemic that was similar to studies 2 and 3 but which included only the most popular articles of which the central claim covered the health, economic, political or social effects of COVID-19. For this study, which ran over 8 days in June 2020, we recruited 1,130 respondents through Qualtrics. A total of 386 of these respondents was presented with one false/misleading online COVID-19-related article within 72 h of publication (an explanation of the extra 24 h delay compared with studies 1–3 is provided in the Methods). We found that searching online increases the probability that a respondent rates a false/misleading article as true by 0.067 ($P = 0.0452$, Cohen's $D = 0.14$, $n = 772$), or an increase in the likelihood of believing a false/misleading article to be true of 20%, and an increase of 0.26 on a seven-point ordinal scale ($P = 0.0054$, Cohen's $D = 0.14$, $n = 772$).

Taken together, studies 1–4 present consistent evidence across a variety of experimental designs, time periods and topics that SOTEN increased belief in misinformation for the point in time investigated. This search effect is concerning on its own but, to better understand the role of search engines and to inform evidence-based interventions, it is also important to evaluate the mechanism underlying these findings. In the next section, we explore one such possible mechanism—exposure to unreliable information corroborating the initial misinformation that was viewed—for why SOTEN can increase belief in misinformation.

## Unreliable results affect misinformation belief

The theory of data voids suggests that, when individuals search online about misinformation, especially misinformation around breaking or recently published news, search engines may return little credible information, instead placing non-credible information at the top of results[20]. These data voids likely exist for a variety of reasons. Low-quality publishers have been found to use search engine optimization techniques and encourage readers to use specific search queries when searching online by consistently using distinct phrases in their stories and in other media[20]. These terms can guide users to data voids on search engines, where only one point of an unreliable view is represented. Low-quality news sources also often re-use stories from each other, polluting search engine results with other similar non-credible stories. It was previously argued (page 75 of ref. 30) that the media dynamics in the United States (particularly on the right) "tend to reinforce partisan statements, irrespective of their truth". Tripodi[31] shows how Google's search algorithms interact with conservative elite messaging strategies to push audiences towards extreme and, at times, false views. This 'propaganda feedback loop' creates a network of outlets reporting the same misinformation and therefore can flood search engine results with false but seemingly corroborating information. The topics and framing of false/misleading news stories are also often distinct from those covered by mainstream outlets, which could limit the amount of reliable news sources being returned by search engines when searching for information about these stories. Finally, direct fact-checks may be difficult to find given that most false narratives are never fact-checked at all and, for stories that are evaluated by organizations such as Snopes

or PoliticFact, these fact-checks may not be posted in the immediate aftermath of a false article's publication. As a result, it would not be surprising that exposure to unreliable news is particularly prevalent when searching online about recently published misinformation.

To investigate the prevalence and effect of exposure to unreliable information while searching for information online, study 5 combines survey data with digital trace data. In this final randomized controlled trial (between-respondents study), we collected articles using the same article-selection protocol and, as in study 1, asked two different groups of respondents to evaluate the same false/misleading or true articles within 72 h of publication and in the same 24 h window. The treatment group was required to search online using Google before providing their assessment of the article's veracity, whereas the control group was not. For those in the treatment group, we collect the URLs that they visited and the top ten Google search engine results to which they were exposed by means of a custom-made browser plug-in that respondents consented to install. Over this 12 day study, we recruited 1,677 respondents living in the United States through Amazon's Mechanical Turk and presented them with three highly popular articles from mainstream and low-quality sources within 72 h of publication. Over the course of this study, 17 false/misleading articles were evaluated by individuals in the control (877 evaluations from 621 unique respondents) and treatment (608 evaluations from 451 unique respondents) groups. By asking the respondents in both the control and treatment groups to install a custom web extension that collected their web browsing behaviour, we were able to collect digital trace data associated with 73% of evaluations of false/misleading articles in the treatment group and 91% of evaluations of false/misleading articles in the control group. This differential attrition was probably due to technical differences between the extension used by the treatment and control groups, but does not result in any substantively meaningful differences between those who completed the survey across the groups (further analysis, including difference in means testing, is provided in the Methods). We still collected the survey results for all of the respondents regardless of compliance and used these responses for the analyses in Fig. 2b. We excluded non-compliant responses from our analysis only when we analysed the effect of the quality of search engine results. We excluded all non-compliant respondents in these analyses to limit possible selection effects, but these respondents were included in all of the other analyses.

Figure 2a presents the proportion of the treatment group's search queries about true and false/misleading articles that return at least one unreliable news source in their Google search engine results. To assess the reliability of a news source, we used classifications from the NewsGuard service, which provides reliability and trustworthiness scores from journalists available at the time of the study (August 2021). Sites with a score of below 60 are deemed to be unreliable, and those with a score above 60 are deemed to be reliable; a histogram of NewsGuard scores for the majority of online news domains is provided in Supplementary Fig. 1 in Supplementary Information C. Figure 2a shows that search queries about true articles are much less likely to return unreliable news among search results than search queries about false/misleading articles (22.5 percentage point difference, $F = 105.8$, $P < 0.0001$). Only 15% of individuals are exposed to at least one unreliable news link when they search about true articles, whereas 38% of individuals are exposed to at least one unreliable news link when they search about false/misleading news.

Using evaluations from study 5, we measured the effect of searching online on the belief in false articles. Figure 2b presents the treatment effect (encouraged to search online) on the probability of believing misinformation using both a dichotomous outcome (rating a false/misleading story as true: 1, yes; 0, no), a seven-point ordinal scale of veracity and a four-point ordinal scale. Like the previous four studies, we found that those who search online about misinformation were more likely to believe false news stories to be true than those who did

not. We found that the effect of SOTEN is greater than in the previous studies, which we suspect may be due to the fact that the search treatment is likely stronger in this study relative to the others given that we could verify compliance for full compensation. In this final study, searching online increased the probability that a respondent rated a false or misleading article as true by 0.107 ($P = 0.0143$, Cohen's $D = 0.21$, $n = 1,485$). Searching online also increased the average score by 0.16 ($P = 0.0434$, Cohen's $D = 0.16$, $n = 1,485$) on a four-point ordinal scale, but not on a seven-point ordinal scale ($P = 0.201$, Cohen's $D = 0.10$, $n = 1,485$). We present the differential effect of SOTEN by political ideology in Extended Data Fig. 2 (explanation for how the ideology of each respondent and the ideological perspective of each article is measured can be found in Supplementary Information I and L).

Using digital trace data collected through the custom browser plug-in, we are able to measure the effect of SOTEN on belief in misinformation among those exposed to unreliable and reliable information by search engines. To this end, we measured the effect of being encouraged to search online on the belief in misinformation for our control group and two subsets of the treatment group: those who were exposed to Google search engine results that returned unreliable results (defined as at least 10% of links coming from news sources with a NewsGuard score below 60) or very reliable results (defined as the first ten links coming only from sources with a NewsGuard score above 85). Roughly 42% of all evaluations in the treatment group fit in either of these two subsets; although subsetting the data in this way ignores 58% of the treatment group, we are interested in the effect of search among groups exposed to very different levels of information quality. Our next analysis looks at the whole set of responses. Across these two subsets, Fig. 2c shows that the probability that an individual believes false/misleading news stories to be true is substantially higher in the treatment group than the control group among respondents whose news exposure is composed of at least 10% unreliable news sites ($n = 1,027$, $P = 0.0035$, Cohen's $D = 0.29$), but it is not higher among those in the treatment group who are only exposed to very reliable news sites ($n = 940$, $P = 0.926$, Cohen's $D = 0.01$) (we confirmed this null result using a Bayesian independent samples $t$-test (interquartile range = 0.707, $BF_{10} = 0.147$) in favour of the null hypothesis). These results are consistent with the theory that lower-quality search engine results can increase belief in misinformation by returning low-quality results. As further evidence, in Fig. 2d, we used the entire sample and calculated the probability of rating misinformation as true by quartile of the mean news quality across the top ten links returned by Google during the evaluation, leading to similar results. Figure 2d shows that respondents who are exposed to search engine results with the lowest-quality news are more likely to rate false/misleading news as true ($n = 1,006$, $P = 0.0241$) compared with those who are not asked to search, whereas those who are exposed to the highest-quality news are not more likely to rate a false/misleading article as true than those in the control group ($n = 1,008$, $P = 0.420$) (we confirmed this null result using a Bayesian independent samples $t$-test (interquartile range = 0.707, $BF_{10} = 0.182$) in favour of the null hypothesis). This again suggests that exposure to unreliable news may explain why SOTEN increases belief in misinformation. Moreover, we found that respondents who are exposed to the top half of information quality in our sample information (top 50%) are no more likely to believe misinformation than those in the control group ($n = 1,113$, $P = 0.429$) (we confirmed this null finding using a Bayesian independent samples $t$-test (interquartile range = 0.707, $BF_{10} = 0.103$) in favour of the null hypothesis). To be clear, the information returned by the Google search results is post-treatment, so this analysis does not infer a causal relationship[32], but it provides evidence consistent with the theory that low-quality information returned by search engines could explain the search effect that we identify. Note that we did not find a statistically significant differential effect of low-quality information on belief across different levels of ideological congruence to the news article (Extended Data Fig. 3).

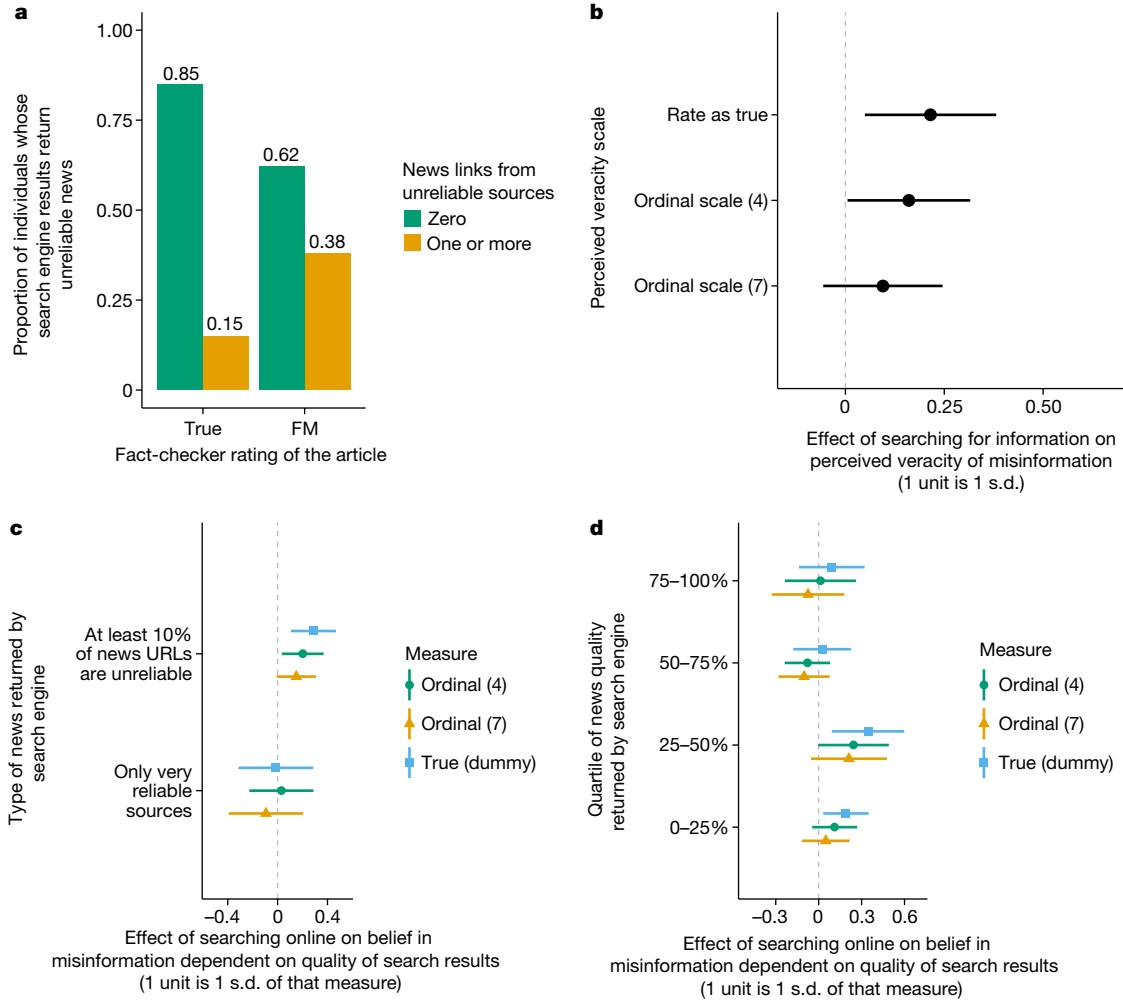

**Fig. 2 | How Google search results impact belief in misinformation (study 5).** **a**, The proportion of individuals who, when searching online about a false/misleading (FM) or true article, are exposed to different levels of unreliable news sites in Google search results. **b**, The average treatment effects and 95% confidence intervals for linear regression models measuring the effect of searching online during study 5 (*n* = 1,485) as a unit of the standard deviation of the dependent variable. Searching online increased the probability that a respondent rated a false/misleading article as true (*P* = 0.0143). **c**, **d**, The same average treatment effects and 95% confidence intervals, but the treatment group was subset by the quality of news returned in their search engine results. **c**, The probability that an individual rates misinformation as true is higher in the treatment group compared with the control group among respondents whose exposure consisted of at least 10% unreliable news sites (*n* = 1,027, *P* = 0.004).

The probability that an individual rates a false/misleading article as true is not different in the treatment group compared to the control group among respondents who were exposed to only very reliable news (*n* = 940, *P* = 0.927). **d**, The probability that an individual rates a false/misleading article as true in the treatment group compared with the control group among respondents who were exposed to the lowest quartile of news quality (*n* = 1,006, *P* = 0.0241) and the second-lowest quartile of news quality (*n* = 1,005, *P* = 0.0116). The probability that an individual rates a false/misleading article as true is not different in the treatment group compared to the control group among respondents who were exposed to the second-highest quartile of news quality (*n* = 1,006, *P* = 0.801) and the highest quartile of news quality (*n* = 1,008, *P* = 0.420). All effects were estimated using OLS with article fixed effects and standard errors clustered at the individual and article level.

If our proposed explanation is indeed correct and exposure to low-quality search results is associated with belief in misinformation, it remains unclear why certain individuals are exposed to low-quality news sources whereas others are not. In the next section, we investigate the search terms that individuals use to see whether this is associated with exposure to low-quality results. Specifically, we consider whether evidence from our study is consistent with two plausible interpretations for why individuals use search terms that are more likely to return low-quality information: ideological congruence with the perspective of the misinformation and digital literacy.

## Individuals exposed to unreliable results

In this section, we assess the viability of two possible explanations for why individuals are exposed to low-quality news in their search

results: (1) ideological congruence and (2) low levels of digital literacy. In the ideological congruence account, partisans may seek out, either consciously or not, information from ideologically congruent sources through the use of search terms that reflect their ideological perspective[33]. Relatedly, although research shows that the most common form of personalization is location-based personalization[34], search engine results for political search queries can be personalized to individual-level characteristics and so the user's ideology may lead to more information that aligns with their ideological worldview[5], possibly amplifying the impact of ideological congruence[6]. This may lead to a concentrated exposure among those ideologically congruent to the misinformation about which they are searching. To this end, we investigated whether exposure to low-quality search results is concentrated among respondents whose self-reported ideology aligns with the ideological slant of the misinformation. Another possible

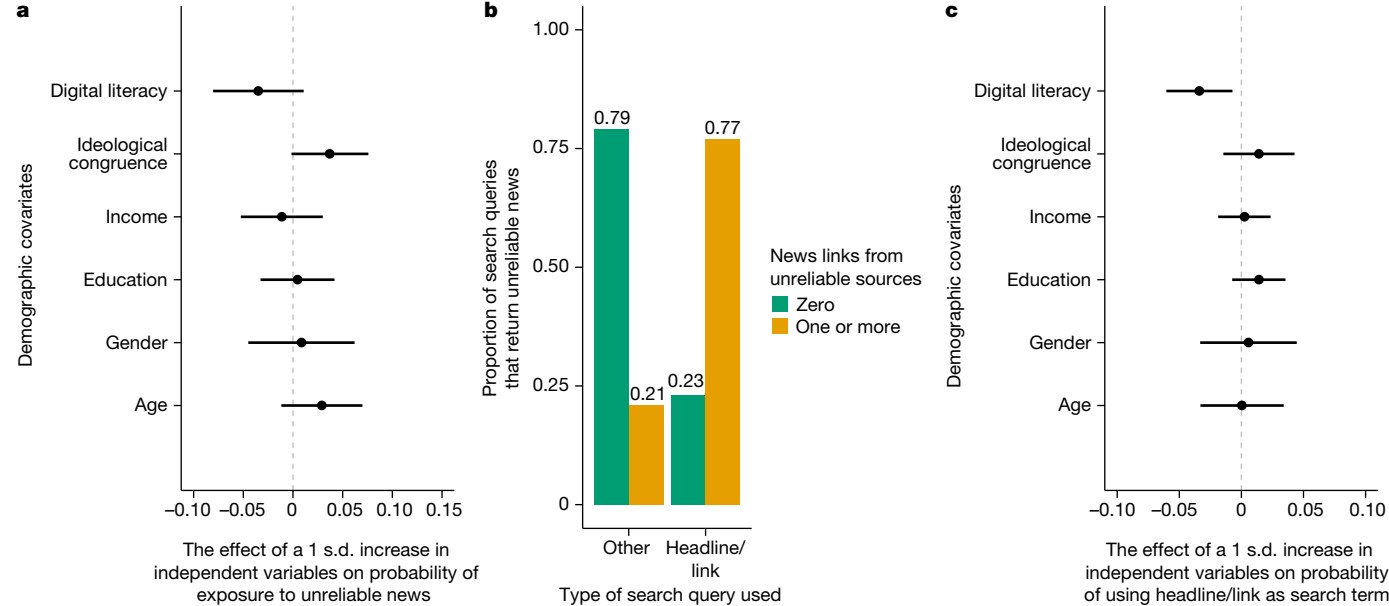

**Fig. 3 | Analysis of the individuals who were exposed to unreliable news sites when evaluating misinformation online (study 5). a**, The effect of demographic variables on the probability of exposure to unreliable news sources when searching online about false/misleading news articles and the 95% confidence intervals during study 5 ($n = 501$). **b**, The proportion of Google searches by individuals ($n = 930$) that return varying numbers of unreliable news sites, when searching online about a false/misleading article. We present these proportions for individuals who used the headline of the article or the link of the article and those who used another query. **c**, The effect of demographic variables on the probability of using the headline/lede or unique URL when searching online about false/misleading news articles and the 95% confidence intervals during study 5 ($n = 930$). Those with lower levels of digital literacy are more likely to use the headline or the unique URL of the false article as their search query when SOTEN, conditioning on ideological congruence and demographics. All effects were estimated using OLS with article fixed effects and standard errors were clustered at the individual and article level.

explanation is that individuals with low levels of digital literacy are more likely to fall into these data voids. Previous research has found that individuals with higher levels of digital literacy use better online information-searching strategies[35], suggesting that those with lower levels of digital literacy may be more likely to use search terms that lead to exposure to low-quality search results. To assess the empirical support for these two potential explanations, we begin by investigating which individual-level characteristics are associated with exposure to unreliable news by fitting an OLS regression model with article-level fixed effects and standard errors clustered at the respondent and article level to predict exposure to unreliable news sites in the search results. We include basic demographic characteristics (income, education, gender and age) in the model. Evidence from these results suggest that lower levels of digital literacy correlate with exposure to unreliable news in search results after conditioning on demographic characteristics. A standard deviation increase in ideological congruence also appears to increase the probability of being exposed to unreliable news by a Google search engine by 0.037 ($P = 0.0827$, Cohen's $D = 0.08$, $n = 501$).

Individuals with lower levels of digital literacy may be more likely to be exposed to unreliable information due to what they actually type into the search engines. To investigate the effect of search terms on the reliability of news returned by the Google search engine, we collected all of the search terms used by individuals in the treatment group. The data-voids theory supposes that, if one uses search terms unique to misinformation, one is more likely to be exposed to low-quality information. To determine whether this affects the quality of search engine results, we coded all search terms for whether they contained the headline or URL of the false article. We found that this is indeed the case. Approximately 9% of all search queries that individuals entered were the exact headline or URL of the original article, and Fig. 3b shows that those who use the headline/lede or the unique URL of misinformation as a search query are much more likely to be exposed to unreliable information in the Google search results. A total of 77% of search queries that used the headline or URL of a false/misleading article as a

search query return at least one unreliable news link among the top ten results, whereas only 21% of search queries that do not use the article's headline or URL return at least one unreliable news link among the top ten results (55.8 percentage point difference, $F = 157.8$, $P < 0.0001$). We run this same analysis excluding the original article from the search engine results and the effect holds. When excluding the original article from the search engine results, 57% of search queries that used the headline or URL of a false/misleading article as a search query return at least one unreliable news link among the top ten results, whereas only 18% of search queries that do not use the article's headline or URL return at least one unreliable news link among the top ten results (39.7 percentage point difference, $F = 85.5$, $P < 0.0001$). The results for all of the relevant figures in the main text excluding the original article from search results are provided in Extended Data Figs. 4 and 5.

To determine who is most likely to use headlines or URLs as their search query, we fit an OLS regression model with article-level fixed effects and standard errors clustered at the respondent and article level to predict using the headline or URL as a search term, again conditioning on basic demographic characteristics. A standard deviation increase in digital literacy decreases the probability of using the headline or the unique URL of the false article as their search query by 0.034 ($P = 0.016$, Cohen's $D = 0.11$, $n = 930$).

Using the headline/lede as a search query probably produces unreliable results because they contain distinct phrases that only producers of unreliable information use[20]. Previous research found that manipulators create content that dominates the search engine environment for people who use certain search terms. An investigation of one article in study 5 appears to support this line of reasoning. Specifically, we analysed the search terms for those searching online about the false/misleading article titled: "U.S. faces engineered famine as COVID lockdowns and vax mandates could lead to widespread hunger, unrest this winter". The term 'engineered famine' in the article is a unique term that is unlikely to be used by reliable sources. An analysis of respondents' search results found that adding the word 'engineered' in front

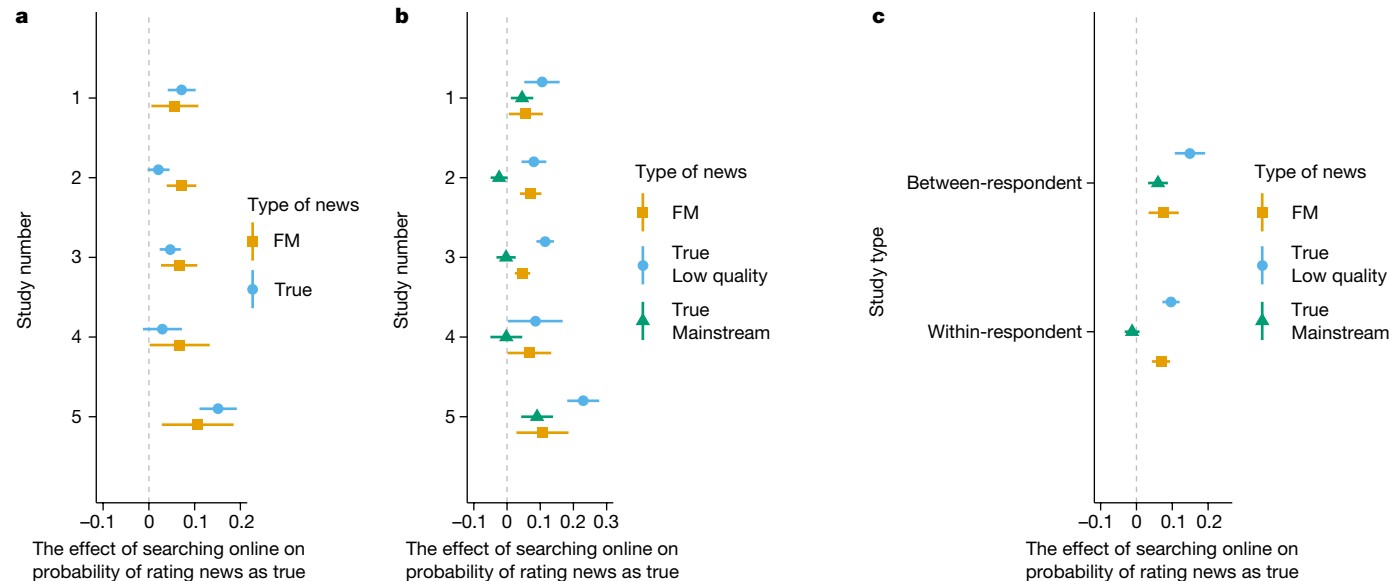

**Fig. 4 | The effect of SOTEN on belief in false/misleading and true news.**
**a**, The effect of searching online on whether individuals rate true news as true and false/misleading news as true and the 95% confidence intervals during studies 1 ($n = 6,269, n = 2,275$), 2 ($n = 6,046, n = 2,020$), 3 ($n = 5,098, n = 1,964$), 4 ($n = 1,420, n = 772$) and 5 ($n = 3,141, n = 1,485$). **b**, The effect of searching online on whether individuals rate true news as true from low-quality sources, true news as true from mainstream sources and false/misleading news as true, and the 95% confidence intervals during studies 1 ($n = 2,782, n = 3,487, n = 2,275$), 2 ($n = 2,596, n = 3,450, n = 2,020$), 3 ($n = 2,490, n = 3,418, n = 1,964$), 4 ($n = 516,$

$n = 904, n = 772$) and 5 ($n = 1,350, n = 1,791, n = 1,485$). **c**, The effect of searching online on whether individuals rate true news as true from low-quality sources, true news as true from mainstream sources and false/misleading news as true, and the 95% confidence intervals for between-respondent experiments (studies 1 and 5) ($n = 4,132, n = 5,278, n = 4,756$) and within-respondent experiments (studies 2–4) ($n = 5,602, n = 7,702, n = 3,760$). All effects were estimated using OLS with article fixed effects and standard errors were clustered at the individual and article level.

of 'famine' changes the search results returned. 0% of search terms that contained the word 'famine' without 'engineered' in front of it returned unreliable results, whereas 63% of search queries that added 'engineered' in front of the word 'famine' were exposed to at least one unreliable result. In fact, 83% of all search terms that returned an unreliable result contained the term 'engineered famine'. See Supplementary Tables 90–94 in Supplementary Information P for data about all searches by respondents about this article, including respondent-level ideology and digital literacy.

## Search effect on belief in true news

Although the finding that SOTEN increases belief in misinformation is concerning in isolation, to fully evaluate the effect of recommending individuals to search online, we must also measure the search effect on belief in true news. We preregistered the hypothesis that searching online would also increase belief in true news and find support for this hypothesis in studies 1–5. For study 1, Fig. 4a shows that searching online increases the probability of correctly rating true news as true by 0.072 ($P = 0.0001$, Cohen's $D = 0.146, n = 6,269$), which is in the same direction as the effect on rating false/misleading as true (0.057; $P = 0.037$, Cohen's $D = 0.12, n = 2,275$). In study 2, in which we set out to test whether the search effect was strong enough to change an individual's evaluation after they had already assessed the veracity of a news story, we found that searching online increases the probability of correctly rating true news as true by only 0.0212 ($P = 0.083$, Cohen's $D = 0.044, n = 6,046$). In study 3, a within-respondent study run months after publication of the articles, searching online increased the probability of correctly rating true news as true by 0.047 ($P = 0.0001$, Cohen's $D = 0.097, n = 5,908$). In study 4, a within-respondent study run strictly on articles about COVID-19, there was no statistically significant search effect on the probability of correctly rating true news as true (0.03, $P = 0.165$, Cohen's $D = 0.062, n = 1,420$) (we confirmed this null finding using a Bayesian independent samples $t$-test (interquartile range = 0.707, $BF_{10} = 0.117$)

in favour of the null hypothesis). There was a large search effect on the probability of rating false/misleading news as true in the same study (0.067; $P = 0.0452$, Cohen's $D = 0.21, n = 772$). In our final study (study 5), a between-respondent experiment with a strict measure of compliance, we found that the search effect on the probability of rating true news as true was significant and in the same direction as the effect identified in study 1, another between-respondent experiment. In study 5, searching online increased the probability of correctly rating true news as true by 0.15 ($P < 0.0001$, Cohen's $D = 0.357, n = 3,141$). These results, as displayed in Fig. 4a, show that the search effect on belief in true news is similar to the search effect on belief in false/misleading news when individuals search online before they determine the veracity of true news (between respondents design), but is smaller or (at times) non-existent when individuals are asked to evaluate true news after having already evaluated the veracity of the true news article (within-respondent design).

Measuring the search effect on all true news ignores that SOTEN may have heterogeneous effects depending on the quality of the source. The source of online news can affect whether an individual believes an article[36] due to a source's reputation[37,38] or the design of the website[39,40]. It is possible that individuals may be less likely to change their perceived veracity of true news from credible sources after searching online if the source's credibility heuristics are relatively strong. However, without receiving a strong signal of source credibility, people may be more likely to believe a true article from a low-quality source if a search engine returns similar coverage from other sources. To this end, we also subset our measurement of the search effect on true articles from mainstream (more reputable) and low-quality (less reputable) sources.

This exploratory analysis shows that the effect of SOTEN for a true article is larger if the article is published by a lower-quality source than if published by a mainstream source in between-respondent experiments ($n = 9,410, P < 0.0001$) and within-respondent experiments ($n = 13,374, P < 0.0001$). In fact, in four out of the five studies, there is only a small or non-existent search effect on the probability of rating true news as

true from mainstream sources. Figure 4b shows that, in study 1, the effect of searching online is significant and in the same direction as for false/misleading news and true news from both low-quality and mainstream sources. Searching online increased the probability of rating true news from mainstream sources as true by 0.045 ($P = 0.0168$, Cohen's $D = 0.10$, $n = 3,487$), increased the probability of rating true news from low-quality sources as true by 0.105 ($P = 0.001$, Cohen's $D = 0.21$, $n = 2,782$) and increased the probability of rating false/misleading news as true by 0.057 ($P = 0.037$, Cohen's $D = 0.12$, $n = 2,275$). When we turn to within-respondent experiments, we find a divergence in the search effect among true news from low-quality sources and true news from mainstream sources. Figure 4b shows that, in study 2, searching online increases the probability of correctly rating true news from low-quality sources as true by 0.081 ($P < 0.0001$, Cohen's $D = 0.16$, $n = 2,596$) but, contrary to study 1, we find that searching online decreases belief in true news from mainstream sources by 0.024 ($P = 0.069$, Cohen's $D = 0.05$, $n = 3,450$). For study 3, Fig. 4b shows that searching online increases the probability that a respondent rates a true article from a low-quality source as true by 0.115 ($P < 0.0001$, Cohen's $D = 0.25$, $n = 2,490$), an effect almost twice the size of the online search effect on false/misleading news in study 3, but that there was no search effect on the probability that a respondent rates a true article from a mainstream source as true ($P = 0.84$, Cohen's $D = 0.01$, $n = 3,418$) (we confirmed this null result using a Bayesian independent samples $t$-test (interquartile range = 0.707, $BF_{10} = 0.039$) in favour of the null hypothesis). The divergence in statistical significance across mainstream and low-quality articles is mirrored in study 4: Fig. 4b shows that searching online increases the probability that a respondent rates a true article from a low-quality source as true by 0.085 ($P = 0.044$, Cohen's $D = 0.17$, $n = 516$), but there was no increase in the probability that a respondent correctly rates a true mainstream story as true ($P = 0.92$, Cohen's $D = 0.01$, $n = 904$) (we confirmed this null result using a Bayesian independent samples $t$-test (interquartile range = 0.707, $BF_{10} = 0.075$) in favour of the null hypothesis). Finally, study 5, a between-respondent experiment with a stronger incentive to search, presents effects with similar direction and significance to the results in study 1. Figure 4b shows that searching online increases the probability that a respondent rates a true article from a low-quality source as true by 0.23 ($P < 0.0001$, Cohen's $D = 0.50$, $n = 1,350$) and increases the probability that a respondent correctly rates a true mainstream story as true by 0.091 ($P = 0.0008$, Cohen's $D = 0.24$, $n = 1,791$).

The results in Fig. 4b also show that there is a possible difference in the search effect on true news from low-quality and mainstream sources, especially in the three within-respondent experiments. To further demonstrate this difference in between-respondent and within-respondent experiments, Fig. 4c presents the search effect when we pool all evaluations of true news (from low-quality and mainstream sources) and false/misleading news articles by experiment type (within-respondent and between-respondent) and re-run the same analysis used to produce the effect sizes in Fig. 4a. In between-respondent experiments, the search effect on belief in true news from mainstream sources is similar to that of false/misleading articles, while the search effect on belief in true news from low-quality sources is larger than the others. In within-respondent experiments, we do not find any search effect on belief in true news from mainstream sources, and the search effect on belief in true news from low-quality sources is significant and in the same direction as the search effect on belief in false/misleading articles. When we substitute the seven-point ordinal scale for the categorical measure, similar results are reported (Extended Data Fig. 6). The results presented in Fig. 4 show that the effect of online search on true news is much larger if the article is published by a low-quality source than if published by a mainstream source in between-respondent experiments ($n = 9,410$, $P < 0.0001$) and within-respondent experiments ($n = 13,374$, $P < 0.0001$). In fact, the effect of SOTEN about a true story from a low-quality source is often similar to or even surpasses the search effect for false articles, and

the effect of SOTEN for true news from mainstream sources is either small or non-existent. It is possible that we do not measure much of an effect of SOTEN on belief in true news from mainstream sources owing to a ceiling effect, as many of our respondents in the control group (those who were not encouraged to search) already rate true news from mainstream sources correctly as true (between 65–80% across all five studies). Taken together, these heterogeneous effects across false and true news articles paint a comprehensive and complex picture of the online search effect.

## Discussion

Across five studies, we found that the act of SOTEN can increase belief in highly popular misinformation by measurable amounts. This result is consistent and robust across five different experimental contexts for the point in time investigated. To better understand the effect of SOTEN and identify potential remedies, we assessed the relative importance of the quality of information returned by search engines in increasing belief in misinformation. Using digital trace data, we provide evidence consistent with the existence of data voids insofar as we find that, when individuals search online about misinformation, they are more likely to be exposed to lower-quality information than when individuals search about true news. Importantly, this exposure may matter: those who are exposed to low-quality information are more likely to believe false/misleading news stories to be true relative to those who are not. Finally, we found evidence that SOTEN increases belief in true news from low-quality sources, but inconsistent evidence of the effect of SOTEN on belief in true news from mainstream sources. The implications of these heterogeneous effects across article veracity and source quality will depend on how people use search engines (that is, the prevalence of searching about false or true news). While practitioners and policymakers must balance the heterogeneous effects of SOTEN across article veracity and source quality, we think that the increase in belief in misinformation should be of particular importance when designing digital media literacy interventions that recommend search as a potential strategy. To be clear, there is a related dynamic that is worthwhile to study, but is not fully captured in this design: namely, online users have full discretion, often without encouragement, around which stories or topics to evaluate through online search. While this process should be the subject of future research that builds on what we have learned here, it is the case that our current study captures the impact of the intended effect of search-based interventions. Specifically, the interventions previously cited[12–14] aim to expand the use of online search engines to evaluate the veracity of news, with the explicit goal of reducing belief in misinformation. However, the impact of search has yet to be established and, therefore, while our design does not perfectly capture the effect of disseminating this recommendation 'in the wild', our results indicate the probable effect of the simple intervention if it were adopted. It should also be noted that a number of media literacy education programmes, such as the Civic Online Reasoning curriculum, provide a larger set of instructions in addition to the search recommendation; however, given the prevalence of the search recommendation across media literacy interventions and the ease with which people can adopt the recommendation, we think that it is important to understand the effect of online search with limited guidance. While our preregistered analysis focuses on the treatment groups who were encouraged to search, we also performed exploratory analysis using control group data that more closely speak to the search effect when people have full discretion over what to search. Using these data, we find a similar effect: people who, without encouragement, searched to evaluate misinformation were more likely to believe it (Extended Data Fig. 7). Future studies could consider using observational data to measure the behavioural impact of disseminating digital media literacy guides, but we think that a better understanding of the impact of SOTEN is a key first step.

In addition to this limitation, we do not allow individuals to select into the news that they would normally read. Allowing this self-selection in communication studies can be of particular importance, as we would like to determine the effect of search on news articles individuals in our study actually read outside of the laboratory[41]. Indeed, studies that do not allow for this self-selection may not correctly identify the heterogeneity of effects across individuals. In our case, we believe exposing individuals to highly popular articles that are widely circulating on social media in the period of most likely exposure captures at least an important part of the pattern of online news consumption. Individuals on social media are becoming more likely to be exposed to viral news on their social media feeds that no longer solely present individuals with what their friends are sharing. Given this shift in online news consumption patterns, we believe that measuring the search effect on highly popular articles is a strength of our design.

The QAnon movement recommends that people "do the research" themselves[42], which seems like a counter-intuitive strategy for a conspiracy-theory-oriented movement. However, our findings suggest that the strategy of pushing people to verify low-quality information online might paradoxically be even more effective at misinforming them. For those who wish to learn more, they risk falling into data voids—or informational spaces in which there is plenty of corroborating evidence from low-quality sources—when using online search engines, especially if they are doing 'lazy searching' by cutting and pasting a headline or URL. Our findings highlight the need for media literacy efforts combatting the effects of misinformation to ground their recommendations in empirically tested interventions, as well as search engines to invest in solutions to the challenges identified here. For example, recent developments in the space—such as the expansion of teaching lateral reading strategies[43] and Google's warning when no credible information is available for given search queries[44]—are interesting steps in this direction and deserve further testing.

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

## Methods

In all six studies, we received informed consent from all of the participants. We also excluded participants for inattentiveness. The researchers were not blinded to the hypotheses when carrying out the analyses. All experiments were randomized. No statistical methods were used to predetermine sample size.

The preregistration for studies 1 and 2 is available online (https://osf.io/akemx/). The methods that we use for all six studies are based on the analysis outlined by this preregistration. It specified that all analyses would be performed at the level of the individual item (that is, one data point per item per participant) using linear regression with standard errors clustered on the participant. The linear regression was preregistered to have a belief in misinformation dummy variable (1 = false/misleading article rated as 'true'; 0 = article rated as 'false/misleading' or 'could not determine') as the dependent variable and the following independent variables: treatment dummy (1 = treatment group; 0 = control group), education (1 = no high school degree; 2 = high school degree; 3 = associates degree; 4 = bachelors degree; 5 = masters degree; 6 = doctorate degree), age, income (0 = US$0–50,000; 1 = US$50,000–100,000; 2 = US$100,000–150,000; 3 = US$150,000+), gender (1 = self-identify as female; 0 = self-identify as not female) and ideology (−3 = extremely liberal; −2 = liberal; −1 = slightly liberal; 0 = moderate; 1 = slightly conservative; 2 = conservative; 3 = extremely conservative). A full description of our variables used in studies 1–4 and study 5 is provided in Supplementary Information I and J. We also stated that we would repeat the main analysis using seven-point ordinal form (1,: definitely false to 7, definitely true) in addition to our categorical dummy variable. Our key prediction stated that the treatment—encouraging individuals to search online—would increase belief in misinformation, which is the hypothesis tested in this study.

However, such an analysis does not account for the likely heterogenous treatment effect across articles evaluated or whether the respondent was ideologically congruent to the perspective of the article. Given this, we deviated from our preregistered plan on two distinct points: (1) to control for the likely heterogeneity in our treatment effect across articles, we add article fixed effects and cluster the standard errors at the article level[45] in addition to at the individual level; and (2) we replace the ideology variable with a dummy variable that accounts for whether an individual's ideological perspective is congruent with the article's perspective. Given that the congruence of one's ideological perspective with that of the article, and not ideology per se, likely affects belief in misinformation, we think that this is the proper variable to use. Although we deviate from these aspects of the preregistered analysis, the results for studies 1–4 using this preregistered model are provided in Extended Data Fig. 8. The results from these models support the hypothesis even more strongly than the results that we present in the main text of this paper.

### Article-selection process

To distribute a representative sample of highly popular news articles directly after publication to respondents, we created a transparent, replicable and preregistered article-selection process that sourced highly popular false/misleading and true articles from across the ideological spectrum to be evaluated by respondents within 24–48 h of their publication. In study 4 (in which we sent only articles about COVID-19 to respondents), we delayed sending the articles to respondents for an additional 24 h to enable us to receive the assessment from our professional fact-checkers before sending the articles out to respondents. Doing so enabled us to communicate fact-checker assessments to respondents once they had completed their own assessment, therefore reducing the chance of causing medical harm by misinforming a survey participant about the pandemic.

We sourced one article per day from each of the following five news streams: liberal mainstream news domains; conservative mainstream news domains; liberal low-quality news domains; conservative low-quality news domains; and low-quality news domains with no clear political orientation. Each day, we chose the most popular online articles from these five streams that had appeared in the previous 24 h and sent them to respondents who were recruited either through Qualtrics (studies 1–4) or Amazon's Mechanical Turk (study 5). An explanation of our sampling technique on Qualtrics and Mechanical Turk, why we chose the services and why we believe that these results can be generalized is provided in Supplementary Information D. Collecting and distributing the most popular false articles directly after publication is a key innovation that enabled us to measure the effect of SOTEN on belief in misinformation during the period in which people are most likely to consume it. In study 3, we used the same articles used in study 2, but distributed them to respondents 3 to 5 months after publication.

To generate our streams of mainstream news, we collected the top 100 news sites by US consumption identified by Microsoft Research's Project Ratio between 2016 and 2019. To classify these websites as liberal or conservative, we used scores of media partisanship from a previous study[46], which assigns ideological estimates to websites on the basis of the URL-sharing behaviour of social media users: websites with a score of below zero were classified as liberal and those above zero were classified as conservative. The top ten websites in each group (liberal or conservative) by consumption were then chosen to create a liberal mainstream and conservative mainstream news feed. For our low-quality news sources, we relied on the list of low-quality news sources from a previous study[3] that were still active at the start of our study in November 2019. We subsequently classified all low-quality sources into three streams: liberal leaning sources, conservative leaning sources and those with no clear partisan orientation. The list of the sources in all five streams, as well as an explanation for how the ideology for low-quality sources was determined, is provided in Supplementary Information E (Supplementary Tables 67–71).

On each day of studies 1, 2 and 5, we selected the most popular article from the past 24 h. We used CrowdTangle, a content discovery and social monitoring platform that tracks the popularity of URLs on Facebook pages, for the mainstream sources, and RSS feeds, for the low-quality sources, from each of the five streams. We used RSS feeds for the low-quality sources instead of CrowdTangle because the Facebook pages of most low-quality sources had been banned and were therefore not tracked by CrowdTangle. Articles chosen by this algorithm therefore represent the most popular credible and low-quality news from across the ideological spectrum. The number of public Twitter (recently renamed X) posts and public Facebook group posts that contained each article in studies 1, 2 and 3 is provided in Supplementary Tables 72 and 73 in Supplementary Information G. In study 3, we used the same articles used in study 2, but distributed them to respondents 3 to 5 months after publication. In study 4, to test whether this search effect is robust to news stories related to the COVID-19 pandemic, we sampled only the most popular articles of which the central claim covered the health, economic, political or social effects of COVID-19. During study 4 and 5, we also added a list of low-quality news sources known to publish pandemic-related misinformation, which was compiled by NewsGuard.

It is important to note that we are testing the search effect during the time period in which our studies run (from study 1 in late 2019 to study 5 in late 2021). It is possible that, over time, the online information environment may change as the result of new search strategies and/or search algorithms.

### Surveys

In each study, we sent out an online survey that asked respondents a battery of questions related to the daily articles that had been selected by our article-selection protocol, as well as a litany of demographic questions. While they completed the survey within the Qualtrics platform, they viewed the articles directly on the website where they had been

originally published. Respondents evaluated each article using a variety of criteria, the most germane of which was a categorical evaluation question: "What is your assessment of the central claim in the article?" to which respondents could choose from three responses: (1) true; (2) misleading/false; and (3) could not determine. The respondents were also asked to assess the accuracy of the news article on a seven-point ordinal scale ranging from 1 (definitely not true) to 7 (definitely true). In study 5, we also asked the respondents to evaluate articles based on a four-point ordinal scale: "to the best of your knowledge, how accurate is the central claim in the article?" (1) Not at all accurate; (2) not very accurate; (3) somewhat accurate; and (4) Very accurate.

We ran our analyses using both categorical responses and the ordinal scale(s). To assess the reliability and validity of both measures, we predict the rating of an article on a seven-point scale using a dummy variable measuring whether that respondent rated that article as true on the categorical measure using a simple linear regression. We found that, across each study, rating an article as true on average increases the veracity scale rating on average by 2.75 points on the seven-point scale (approximately 1.5 s.d. of the ratings on the ordinal scale). The full results are shown in Extended Data Fig. 9. To ensure that responses that we use were actually from respondents who evaluated articles in good faith, two relatively simple attention checks for each article, which do not depend on any ability associated with the evaluation task, were used. If a respondent failed any of these attention checks, all of their evaluations were omitted from this analysis. These attention check questions can be found in Supplementary Information F.

### Determining the veracity of articles

One of the key challenges in this study was determining the veracity of the article in the period directly after publication. Whereas many studies use source quality as a proxy for article quality, not all articles from suspect news sites are actually false[3]. Other studies have relied on professional fact-checking organizations such as Snopes or Politifact to identify false/misleading stories from these sources[47,48]. However, the use of evaluations from these organization is impossible when sourcing articles in real time because we have no way of knowing whether these articles will ever be checked by such organizations. As an alternative evaluation mechanism, we hired six professional fact checkers from leading national media organizations to also assess each article during the same 24 h period as respondents. In studies 4 and 5, given the onset of the pandemic and the potential harm caused by medical misinformation, the professional fact-checkers rated the articles 24 h before the respondents so that we could show respondents the fact-checkers' ratings of each article immediately after completion of the survey. These professional fact-checkers were recruited from a diverse group of reputable publications (none of the fact-checkers were employed by a publication included in our studies to ensure no conflicts of interest) and were paid US$10.00 per article. The modal response of the professional fact checkers yielded 37 false/misleading, 102 true and 16 indeterminate articles from study 1. Most articles were evaluated by five fact-checkers; a few were evaluated by four or six. A different group of six fact-checkers evaluated all of the articles during studies 4 and 5 relative to studies 1–3. We use the modal response of the professional fact checkers to determine whether we code an article as 'true', 'false/misleading' or 'could not determine'. We are then able to assess the ability of our respondents to identify the veracity of an article by comparing their response to the modal professional fact checker response. In terms of inter-rater reliability among fact-checkers, we report a Fleiss' Kappa score of 0.42 for all fact-checker evaluations of articles used in this paper. We also report the article-level agreement between each pair of fact-checkers and average weighted Cohen kappa score between each pair of fact-checkers in Supplementary Table 74 in Supplementary Information K. These scores are reported for the articles that were rated by five professional fact-checkers. Although this level of agreement is quite low, it is slightly higher than other studies that have used

professional fact-checkers to rate the veracity of both credible and suspect articles using similar scale our fact-checkers used[49]. This low level of agreement of professionals over what is misinformation may also explain why so many respondents believe misinformation and why searching online does not effectively reduce this problem. Identifying misinformation is a difficult task, even for professionals.

We also present all of the analyses in this paper using only false/misleading articles with a robust mode—which we define as any modal response of fact-checkers that would not change if one professional fact-checker changed their response—to remove articles where there was higher levels of disagreement among professional fact-checkers. These results can be found in Supplementary Table 74 Supplementary Information K. We found that the direction of our results does not change when using the false/misleading articles with a robust mode, although the effect is no longer statistically significant for 2 out of the 4 studies using the categorical measure and 1 out of the 4 studies using the continuous measure. To determine whether the search effect changes with the rate of agreement of fact-checkers, we ran an interaction model and present the results in Extended Data Fig. 10. We found that the search effect does appear to weaken for articles that fact-checkers most agree are false/misleading. Put another way, the search effect is strongest for articles in which there is less fact-checker agreement that the article is false, suggesting that online search may be especially ineffective when the veracity of articles is most difficult to ascertain. Although this is the case, the search effect for only false/misleading articles with a robust mode (one fact-checker changing their decision from false/misleading to true will not change the modal fact-checker evaluation) is still quite consistent and strong. These results are presented in Supplementary Figs. 2–5 in Supplementary Information M.

### Study 1

In study 1, we tested whether SOTEN affects belief in misinformation in a randomized controlled trial that ran for 10 days. During this study, we asked two different groups of respondents to evaluate the same false/misleading or true articles in the same 24 h window, but asked only one of the groups to do this after searching online. We preregistered a hypothesis that both false/misleading and true news were more likely to be rated as true by those who were encouraged to search online. This study was approved by the New York University Committee on Activities Involving Human Subjects (IRB-FY2019-3511).

### Participants and materials

On ten separate days (21 November 2019 to 7 January 2020), we randomly assigned a group of respondents to be encouraged to search online before providing their assessment of the article's veracity. Over these 10 days, 13 different false/misleading articles were evaluated by individuals in our control group who were not requested to search online (resulting in 1,145 evaluations from 876 unique respondents) and those in our treatment group who were requested to search online (resulting in 1,130 evaluations from 872 unique respondents). The articles used during this study can be found in Supplementary Tables 1–5 in Supplementary Information A.

### Procedure

The participants in both the control and treatment group were given the following instructions at the beginning of the survey: "In this survey you will be asked to evaluate the central claim of three recent news articles". We then presented the participants with three out of five articles selected that day randomly (no articles could be shown to a respondent more than once). For each article, the respondents in each group were asked a series questions about the article, such as whether it is an opinion article, their interest in the article, and their perceived reliability of the source. Those in the control group were presented with the veracity questions most relevant to this study: "What is your assessment of the central claim in the article?" with the following options:

(1) true: the central claim you are evaluating is factually accurate. (2) Misleading and/or false: misleading: the central claim takes out of context, misrepresents or omits evidence. False: the central claim is factually inaccurate. (3) Could not determine: you do not feel you can judge whether the central claim is true, false or misleading. The participants were also asked a seven-point ordinal scale veracity question: "now that you have evaluated the article, we are interested in the strength of your opinion. Please rank the article on the following scale: 1 (definitely not true), 2, 3, 4, 5, 6, 7 (definitely true)". Differing from the control group, the participants in the treatment group (encouraged to search for additional information) were given instructions before these two veracity questions (see below). These instructions encouraged them to search online and asked the respondents questions about their search online.

**Instructions to find evidence to evaluate central claim.** The following instructions were provided to respondents in studies 1–5 before SOTEN.

"The purpose of this section is to find evidence from another source regarding the central claim that you're evaluating. This evidence should allow you to assess whether the central claim is true, false or somewhere in between. Guidance for the finding evidence for or against the central claim you've identified:

(1) By evidence, we mean an article, statement, photo, video, audio or statistic relevant to the central claim. This evidence should be reported by some other source than the author of the article you are investigating. This evidence can either support the initial claim or go against it.

(2) To find evidence about the claim, you should use a keyword search on a search engine of your choice or within the website of a particular source you trust as an authority on the topic related to the claim you're evaluating.

(3) We ask that you use the highest-quality pieces of evidence to evaluate the central claim in your search. If you cannot find evidence about the claim from a source that you trust, you should try to find the most relevant evidence about the claim you can find from any source, even one you don't trust.

For additional instructions explaining how to find evidence please click this text" (these additional instructions are provided in Supplementary Information H, and the instructions that we gave respondents for the extra study omitting some instructions are provided in Supplementary Information O).

We next presented respondents with the following four questions:

(1) What are the keywords you used to research this original claim? If you searched multiple times, enter just the keywords you used on your final/successful search. If you used a reverse image search, please enter "reverse image search" in the text box.

(2) Which of the following best describes the highest quality evidence you found about the claim in your search? Possible responses: (A) I found evidence from a source that I trust. (B) I found evidence, but it's from a source that I don't know enough about to trust or distrust. (C) I found evidence, but it's from a source that I don't trust. (D) I did not find evidence about this claim.

(3) Evidence link: please paste the link for the highest quality evidence you found (paste only the text of the URL link here. Do not include additional text from the webpage/article, etc.). If you did not find any evidence, please type the following phrase in the text box below: "No Evidence".

(4) Additional evidence links: if you use other different evidence sources that are particularly helpful, please paste the additional sources here.

After the participants read the instructions and were asked these questions about their online search, those in the treatment group were presented with the two veracity questions of interest (categorical and seven-point ordinal scale). In both the control and treatment conditions, the response options were listed in the same order as they are listed in this section.

**Analysis plan**
This analysis was preregistered (https://osf.io/akemx/).

**Balance table.** Supplementary Table 95 in Supplementary Information Q compares basic demographic variables among respondents in the control and treatment group. This table shows that respondents were similar across demographic variables, except for income. Those in the control group self-reported making higher levels of income than those in the treatment group. We did not record the data for 83.2% of those who entered the survey and were in the control group and 85.8% of those in the treatment group. The majority of respondents dropped out of the survey at the beginning. About 66% of all respondents who entered the survey refused to consent or did not move past the first two consent questions. Taken together, of all of the respondents who moved past the consent questions, 51% of respondents dropped out of the survey in the control group and 58% of the respondents dropped out of the survey in the treatment group. About 11% of those who did not complete the survey did so because they failed the attention checks and were removed from the survey.

**Study 2.** Study 2 ran similarly to study 1, but over 29 days between 18 November 2019 and 6 February 2020. In each survey that was sent in study 1, we asked respondents in the control group to evaluate the third article they received a second time, but only after looking for evidence online (using the same directions to search online that participants in study 1 received).

This study measures the effect of searching online on belief in misinformation but, instead of running a between-respondent random control trial, we run a within-respondent study. In this study, the participants first evaluated articles without being encouraged to search online. After providing their veracity evaluation on both the categorical and ordinal scales, they were encouraged to search online to help them re-evaluate the article's veracity using the same instructions as from study 1. This is probably a more difficult test of the effect of searching online, as individuals have already anchored themselves to their previous response. Literature on confirmation bias leads us to believe that new information will have the largest effect when individuals have not already evaluated the news article on its own. Thus study therefore enables us to measure whether the effect of searching online is strong enough to change an individual's evaluation of a news article after they have evaluated the article on its own. We did not preregister a hypothesis, but we did pose this as an exploratory research question in the registered report for study 1. This study was approved by the New York University Committee on Activities Involving Human Subjects (IRB-FY2019-3511).

**Participants and materials.** During study 2, 33 unique false or misleading articles were evaluated and re-evaluated by 1,054 respondents. We then compared their evaluation before being requested to search online and their evaluation after searching online. The articles used during this experiment are provided in Supplementary Tables 6–12 in Supplementary Information A. Summary statistics for all of the respondents in this study are presented in Supplementary Table 96 in Supplementary Information Q.

**Procedure.** Similar to study 1, respondents initially evaluated articles as if they were in the control group, but after they finished their evaluation they were then presented with this text: "Now that you have evaluated the article, we would like you evaluate the article again, but this time find evidence from another source regarding the central claim that you're evaluating". They were then prompted with the same instructions and questions as the treatment group in study 1.

**Analysis plan.** This analysis was posed as an exploratory research question in the registered report for study 1.

**Study 3.** Although no pre-analysis plan was filed for study 3, this study replicated study 2 using the same materials and procedure, but was run between 16 March 2020 and 28 April 2020, 3–5 months after the publication of each these articles. This study set out to test whether this search effect remained largely the same months after the publication of misinformation when professional fact-checks and other credible reporting on the topic are hopefully more prevalent. This study was approved by the New York University Committee on Activities Involving Human Subjects (IRB-FY2019-3511).

**Participants and materials.** In total, 33 unique false or misleading articles were evaluated and re-evaluated by 1,011 respondents. We then compared their evaluation before being requested to search online and their evaluation after searching online. The articles used during this experiment are provided in Supplementary Tables 6–12 in Supplementary Information A. Summary statistics for all respondents in this study are presented in Supplementary Table 97 in Supplementary Information Q.

**Analysis plan.** No preregistration was filed for this study.

**Study 4.** Although no pre-analysis plan was filed for study 4, this study extended study 2 by asking individuals to evaluate and re-evaluate highly popular misinformation strictly about COVID-19 after searching online. This study was run over 8 days between 28 May 2020 to 22 June 2020. In the 'Article-selection process' section, we describe the changes that we made in our article-selection process to collect these articles. We collected these articles and sent them out to be evaluated by respondents. This study measured whether the effect of searching online on belief in misinformation still holds for misinformation about a salient event, in this case the COVID-19 pandemic. This study was approved by the New York University Committee on Activities Involving Human Subjects (IRB-FY2019-3511). This IRB submission is the same as the one used for studies 1, 2 and 3, but it was modified and approved in May 2020 before we sent out articles related to COVID-19.

**Participants and materials.** A total of 13 false or misleading unique articles was evaluated and re-evaluated by 386 respondents. We then compared their evaluation before being requested to search online (the treatment) and their evaluation after searching online. The articles used during this experiment are provided in Supplementary Tables 13–17 in Supplementary Information A. Summary statistics for all of the respondents in this study are presented in Supplementary Table 98 in Supplementary Information Q.

**Analysis plan.** No preregistration was filed for this study.

**Study 5.** To test the effect of exposure to unreliable news on belief in misinformation, we ran a fifth and final study that combined survey and digital trace data. This study was almost identical to study 1, but we used a custom plug-in to collect digital trace data and encouraged the respondents to specifically search online using Google (our web browser plug-in could collect search results only from a Google search result page). Similar to study 1, we measured the effect of SOTEN on belief in misinformation in a randomized controlled trial that ran on 12 separate days from 13 July 2021 to 9 November 2021, during which we asked two different groups of respondents to evaluate the same false/misleading or true articles in the same 24 h window. The treatment group was encouraged to search online, while the control group was not. This study was approved by the New York University Committee on Activities Involving Human Subjects (IRB-FY2021-5608).

**Participants and materials.** Unlike the other four studies, these respondents were recruited through Amazon Mechanical Turk. Only workers within the United States (verified by IP address) and those with above a 95% success rate were allowed to participate. We were unable to recruit a representative sample of Americans using sampling quotas owing to the difficulty of recruiting respondents from Amazon Mechanical Turk who were willing to install a web-tracking browser extension in the 24 h period after our algorithm selected articles to be evaluated.

Over 12 days during study 5, a group of respondents were encouraged to SOTEN before providing their assessment of the article's veracity (treatment) and another group was not encouraged to search online when they evaluated these articles (control). A total of 17 different false/misleading articles were evaluated by individuals in our control group who were not encouraged to search online (877 evaluations from 621 unique respondents) and those in our treatment group who were encouraged to search online (608 evaluations from 451 unique respondents). The articles used during this experiment are provided in Supplementary Tables 18–22 in Supplementary Information A. We do not find statistically significant evidence that respondents who we were recruited to the control group were different on a number of demographic variables. Supplementary Table 99 in Supplementary Information Q compares those in the treatment and control group. Only 20% of those in the control group who consented to participate in the survey dropped out of the study, whereas 62% of those who entered the survey and were in the treatment group dropped out of the study. This difference in compliance rates can be explained by the difference in the web extension for the treatment group relative to the one given to the control group. For technical reasons related to capturing HTML, the respondents in the treatment group had to wait at least 5 s for the web extension that was installed to collect their Google search engine results, which may have resulted in some respondents accidentally removing the web extension. If they did not wait for 5 s on a Google search results page, the extension would turn off and they would have to turn it back on. These instructions were presented clearly to the respondents, but probably resulted in differences in compliance. This differential attrition does not result in any substantively meaningful differences between those who completed the survey in the treatment and control group as shown in Supplementary Table 99 in Supplementary Information Q.

**Procedure.** The participants in both the control and treatment group were given the following instructions at the beginning of the survey: "In this survey you will be asked to evaluate the central claim of three recent news articles". Those assigned to the treatment group were then asked to install a web extension that would collect their digital trace data including their Google search history. They were presented with the following text: "In this section we will ask you to install our plugin and then evaluate three news articles. To evaluate these news articles we will ask you to search online using Google about each news article online and then use Google Search results to help you evaluate the news articles. We need you to install the web extension and then search on Google for relevant information pertaining to each article in order for us to compensate you". They were then presented with instructions to download and activate the "Search Engine Results Saver", which is available at the Google Chrome store (https://chrome.google.com/webstore/detail/search-engine-results-sav/mjdfiochiimhfgbdgkielodbojlpfcbl?hl=en&authuser=2). Those assigned to the control group were also asked to install a web extension that collected their digital trace data, but not any search engine results. They were presented with the following text: "In this section we will ask you to install our plugin and then evaluate three news articles. You must install the extension, log in and keep this extension on for the whole survey to be fully compensated". They were then presented with instructions to download and activate URL Historian, which is available at the Google Chrome store (https://chrome.google.com/webstore/detail/url-historian/imdfbahhoamgbblienjdoeafphlngdim).

Both those in the control and treatment group were asked to download and install a web extension that tracked their web behaviour to limit varying levels of attrition across both groups, due to the unwillingness or inability of respondents to install this kind of extension. After the respondents downloaded their respective web extension, the study ran identical to study 1.

**Digital trace data.** By asking individuals to download and activate web browsers that collected their URL history and scraped their search engine results, we were able to measure the quality of news they were exposed to when they searched online. We were unable to collect this data if respondents did not search on Google, deactivated their web browser while they were taking the survey, or did not wait on a search engine result page for at least 5 s. Thus, in total for the 653 evaluations of misinformation in our treatment group, we collected Google search results for 508 evaluations (78% of all evaluations). We also collected the URL history of those in the control group, but did not use these data in our analyses. For most demographic characteristics (age, gender, income and education), we have statistically significant evidence that respondents from whom we were able to collect search engine results were slightly different compared with those from whom we were not able to collect these results. We find that participants from whom we were able to collect this digital trace data were more likely to self-identify as liberal by about 0.8 on a seven-point scale, more likely to self-report higher levels of digital literacy and less likely to self-identify as female. Supplementary Table 100 in Supplementary Information Q compares complying and non-complying individuals within the treatment group. Those compliant in the treatment group were slightly younger by two and a half years and slightly more likely to be male.

**Analysis plan.** No preregistration was filed for this study.

When we analysed the effect of the quality of online information, we included only those in the control group who kept their web extension on during the survey to limit possible selection bias effects. In the control group, 93% of the respondents evaluated a false/misleading article in the control group installed the web extension that tracked their own digital trace data throughout the whole survey. Similar to the treatment group, we do find that those for whom we were able to collect this digital trace data were more likely to self-identify as liberal by about 0.55 on a seven-point scale and more likely to self-report higher levels of digital literacy. The magnitude of these differences are modest and the direction of these differences are identical to the differences in the treatment group. Supplementary Table 101 in Supplementary Information Q compares complying and non-complying individuals within the control group. We do not see large differences in how those who are compliant in the control group differ from those who are compliant in the treatment group. Supplementary Table 102 in Supplementary Information Q compares complying individuals in the treatment and control groups.

To measure the quality of search results, we use scores from Newsguard, an internet plug-in that informs users whether a site that they are viewing is reliable. NewsGuard employs a team of trained journalists and experienced editors to review and rate news and information websites based on nine criteria. The criteria assess basic practices of journalistic credibility and transparency, assigning a score from 0 to 100. Sites with a score below 60 are deemed to be unreliable, and those with a score of above 60 are deemed to be reliable. NewsGuard has ratings for over 5,000 online news domains, responsible for about 95% of all the news consumed in the United States, United Kingdom, France, Germany and Italy. More information is available online (https://www.newsguardtech.com). A sample of their ratings can be found online (https://www.newsguardtech.com/ratings/sample-nutrition-labels/). The full list of online news domains and their ratings is licensed by NewsGuard to approved researchers.

**Study 6.** Study 6 tests whether the search effects that we identify on belief in false/misleading and true articles still hold when we change the instructions we present to respondents. To this end, we ran an experiment similar to study 1, but we added two other treatment arms in which we encouraged individuals to search online to evaluate news. This study was approved by the New York University Committee on Activities Involving Human Subjects (IRB-FY2019-3511).

**Ethics.** We complied with all relevant ethical regulations. All of the studies were reviewed and approved by the NYU Institutional Review Board (IRB). Studies 1, 2, 3 and 4 were approved by NYU IRB protocol IRB-FY2019-351. Study 5 was approved by NYU IRB protocol IRB-FY2021-5608. Study 6 was approved by a modified NYU IRB protocol IRB-FY2019-3511. All of the experimental participants provided informed consent before taking part. The participants were given the option to withdraw from the study while the experiment was ongoing as well as to withdraw their data at any time.

### Reporting summary

Further information on research design is available in the Nature Portfolio Reporting Summary linked to this article.

## Data availability

All custom scripts and all data used this study have been made available at GitHub (https://github.com/SMAPPNYU/Do_Your_Own_Research). The preregistrations for studies 1 and 2 are available online (https://osf.io/akemx/). All other relevant data are available from the corresponding author on request. Source data are provided with this paper.

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

**Acknowledgements** We thank M. A. Brown for her help in the production of the Chrome browser extension and data collection in studies 1–5; C. Ji for supporting the development of the browser extension; S. Messing and numerous seminar participants at ETH Zurich, Stanford University, Williams College, Dartmouth College, the Toulouse School of Economics, Georgetown University, and the New York University Center for Social Media and Politics for feedback and comments. We acknowledge project funding from the National Science Foundation (2029610) and the William and Flora Hewlett Foundation. The Center for Social Media and Politics at New York University is supported by funding from the John S. and James L. Knight Foundation, the Charles Koch Foundation, Craig Newmark Philanthropies, the William and Flora Hewlett Foundation, the Siegel Family Endowment and the Bill and Melinda Gates Foundation. This work was supported in part through the NYU IT High Performance Computing resources, services and staff expertise.

**Author contributions** K.A., Z.S., W.G. and J.A.T. designed the research, including the article-selection procedures, survey instruments and analysis plans. K.A., Z.S., W.G., J.A.T., J.N. and N.P. designed the research infrastructure. K.A. and W.G. analysed the data. Z.S. and K.A. wrote the paper. J.A.T. and J.N. revised the paper.

**Competing interests** W.G. is currently an employee at Google. While he contributed to this paper before his employment, he made no contributions after accepting or starting employment at the company. There are no other competing interests, financial or non-financial, to report.

**Additional information**
**Correspondence and requests for materials** should be addressed to Kevin Aslett.

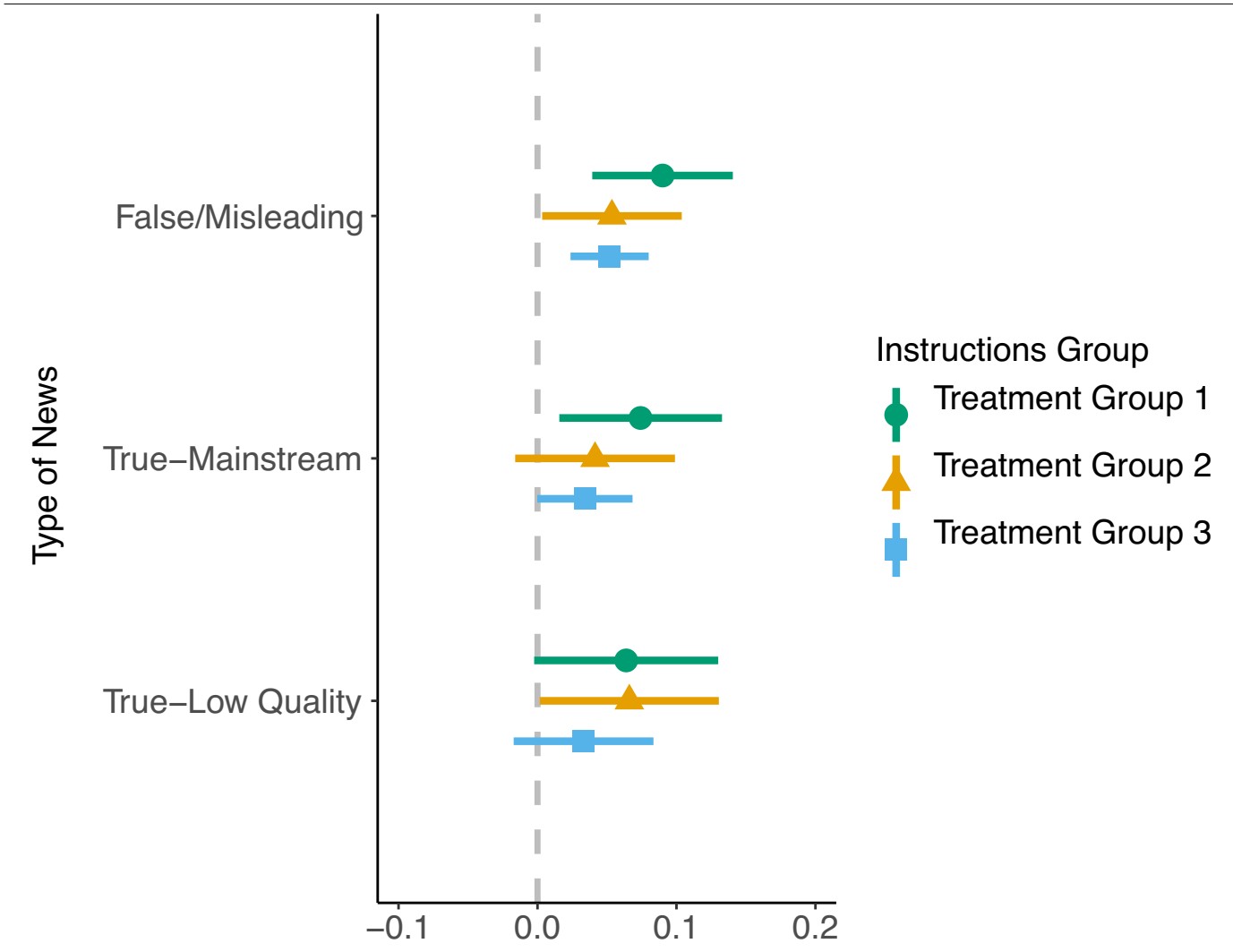

## Effect of Searching Online on Belief in News Dependent on Type of Instructions

**Extended Data Fig. 1 | The online search effect using different online search instructions (categorical veracity measure).** This figure displays the average treatment effect of SOTEN on rating a false/misleading article as true and 95 percent confidence intervals using different online search instructions in Study 6. It shows that the effect of searching online increases the probability of rating a false/misleading article as true regardless of the instructions given to respondents. When comparing the control group ($N = 1,113$) to treatment group 1 ($N = 1,075$; the same instructions used in Studies 1–5), treatment group 2 ($N = 1,034$; limited instructions), and treatment group 3 ($N = 1,036$; no instructions), searching online increased the likelihood of rating false/misleading news as true by 0.09 ($P = 0.0027$), 0.05 ($P = 0.0389$), and 0.05 ($P = 0.0021$) respectively. The effects of online search were similar for true news from mainstream sources and true news from low-quality sources. All effects are estimated using ordinary least squares with article fixed effects and standard errors clustered at the individual and article level.

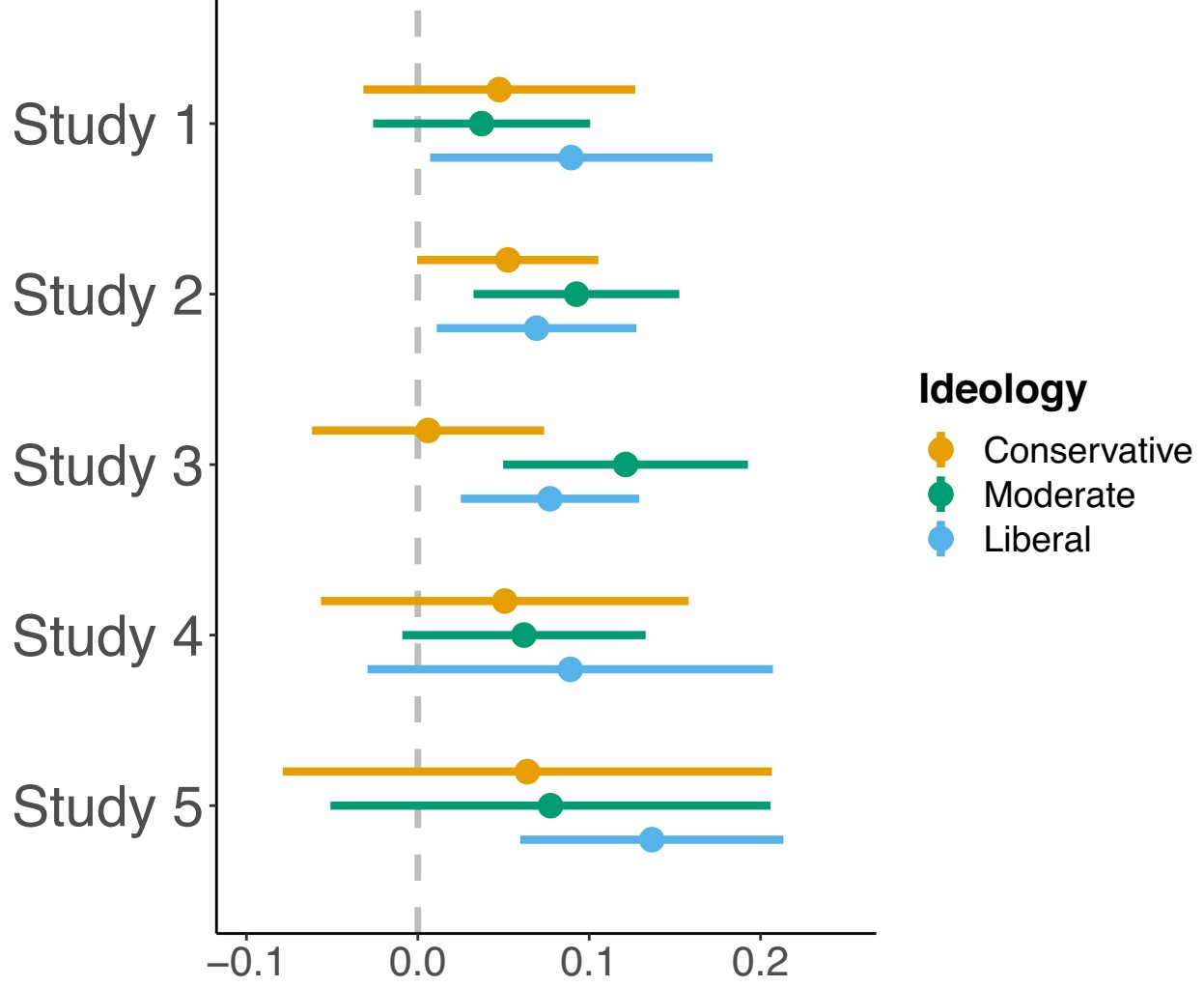

## Effect of Searching Online on Probability
## of Rating Misinformation as True

**Extended Data Fig. 2 | Search effect across self-reported political ideology.**
This figure presents the effect of searching online on rating a false/misleading
article as true and 95 percent confidence intervals subset by political ideology
(Liberal, Moderate, and Conservative) in during Studies 1 (N = 780, N = 712,
N = 783), 2 (N = 664, N = 670, N = 686), 3 (N = 700, N = 594, N = 670), 4 (N = 270,
N = 226, N = 276), 5 (N = 757, N = 465, N = 249). Generally, the effect sizes are
quite similar across political ideological groups. All effects are estimated using
ordinary least squares with article fixed effects and standard errors clustered
at the individual and article level.

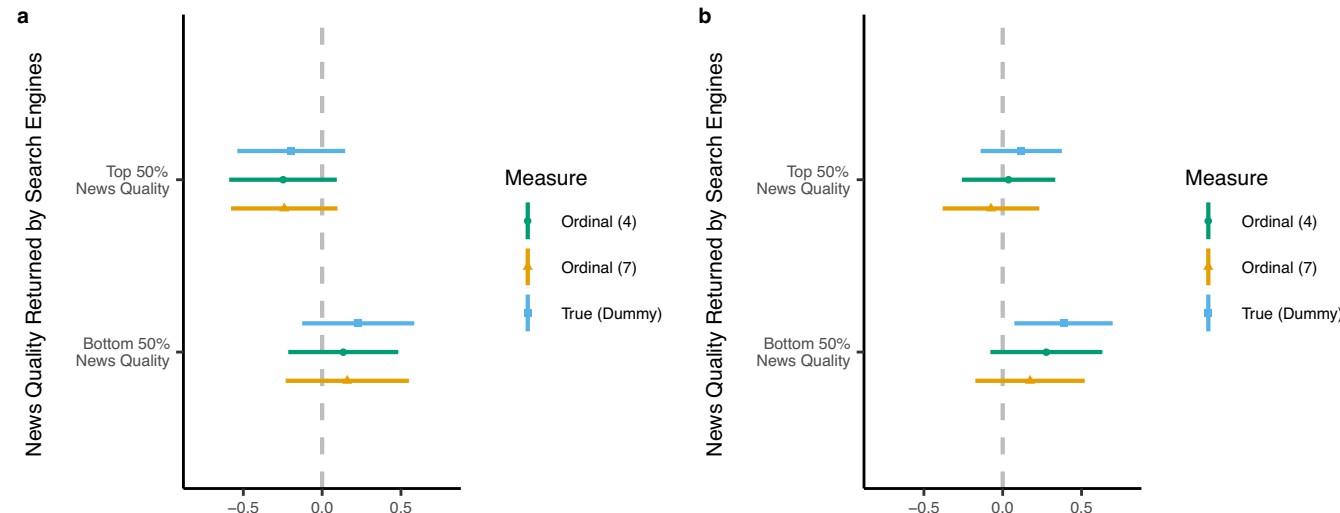

**Extended Data Fig. 3 | Who is most susceptible to unreliable information when searching for more information? (Study 5).** Panels a through b present the effect of searching online on rating a false/misleading article as true and 95 percent confidence intervals during Study 5 as a unit of the standard deviation of the dependent variable. Marginal effects are subset by the quality of news returned in their search engine results (top 50% and bottom 50% of average source quality of news returned). Panel a presents the effect of being encouraged to search online among those ideologically congruent with the ideological perspective of the item of misinformation they are evaluating (N = 562, N = 320), while panel b presents the search effect of being encouraged to search online among those ideologically incongruent with the ideological perspective of the item of misinformation they are evaluating (N = 790, N = 428). All effects are estimated using ordinary least squares with article fixed effects and standard errors clustered at the individual and article level.

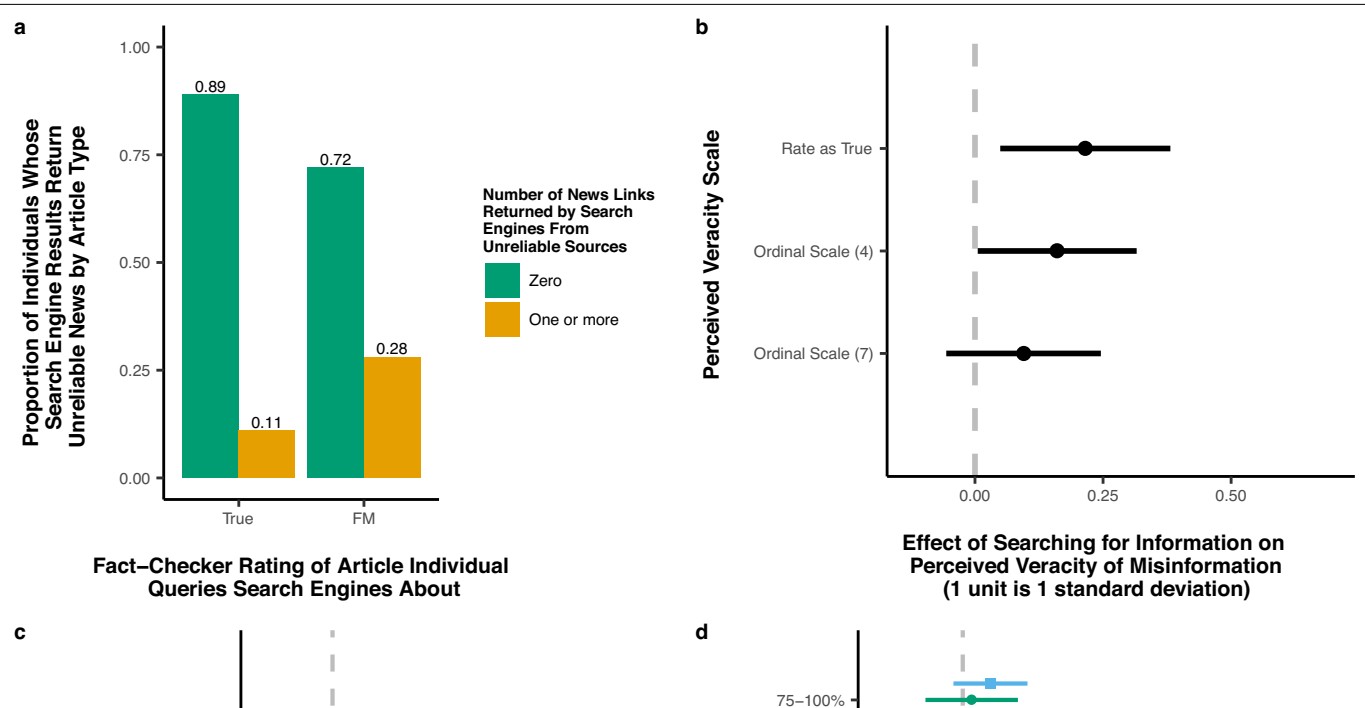

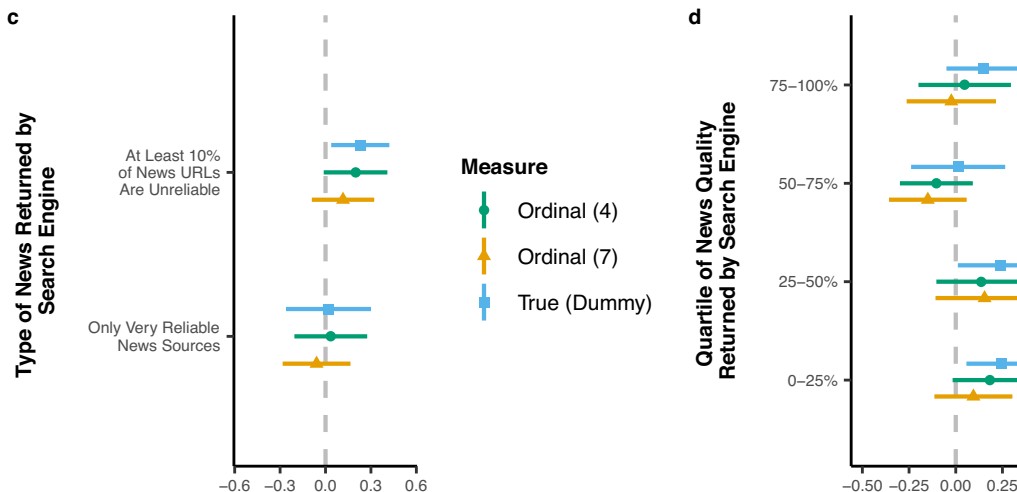

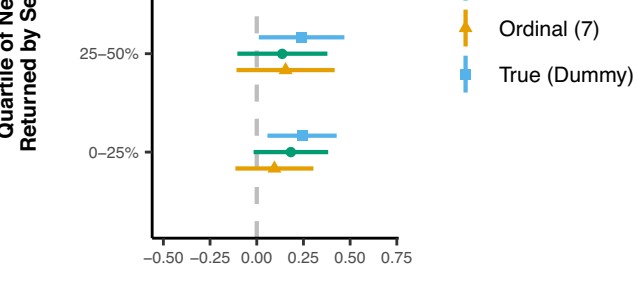

**Extended Data Fig. 4 | How does news returned in Google search results affect belief in misinformation? (Study 5) - excluding original article in search results analysis.** Panel a presents the proportion of individuals who, when searching online about a false/misleading or true article, are exposed to different levels of unreliable news sites in Google search results. Panel b presents the average treatment effects and 95 percent confidence intervals for linear regression models measuring the effect of searching online during Study 5 (N = 1,485) as a unit of the standard deviation of the dependent variable. Searching online increased the probability a respondent rated a false/misleading article as true. Subsetting the treatment group by the quality of news returned in their search engine results, Panel c and d present these same average treatment effects and 95 percent confidence intervals. Panel c shows that the probability an individual rates misinformation as true is higher than the control group

among respondents who are exposed to at least one unreliable news site (N = 986). The probability an individual a false/misleading article as true is not different than the control group among respondents who are exposed to only very reliable news (N = 958). Panel d shows that the probability an individual rates a false/misleading article as true than the control group among respondents who are exposed to the lowest quartile of news quality (N = 1,006) and second lowest quartile of news quality (N = 1,005). The probability an individual rates a false/misleading article as true is not different than the control group among respondents who are exposed to the second highest quartile of news quality (N = 1,005) and the highest quartile of news quality (N = 1,006). All effects are estimated using ordinary least squares with article fixed effects and standard errors clustered at the individual and article level.

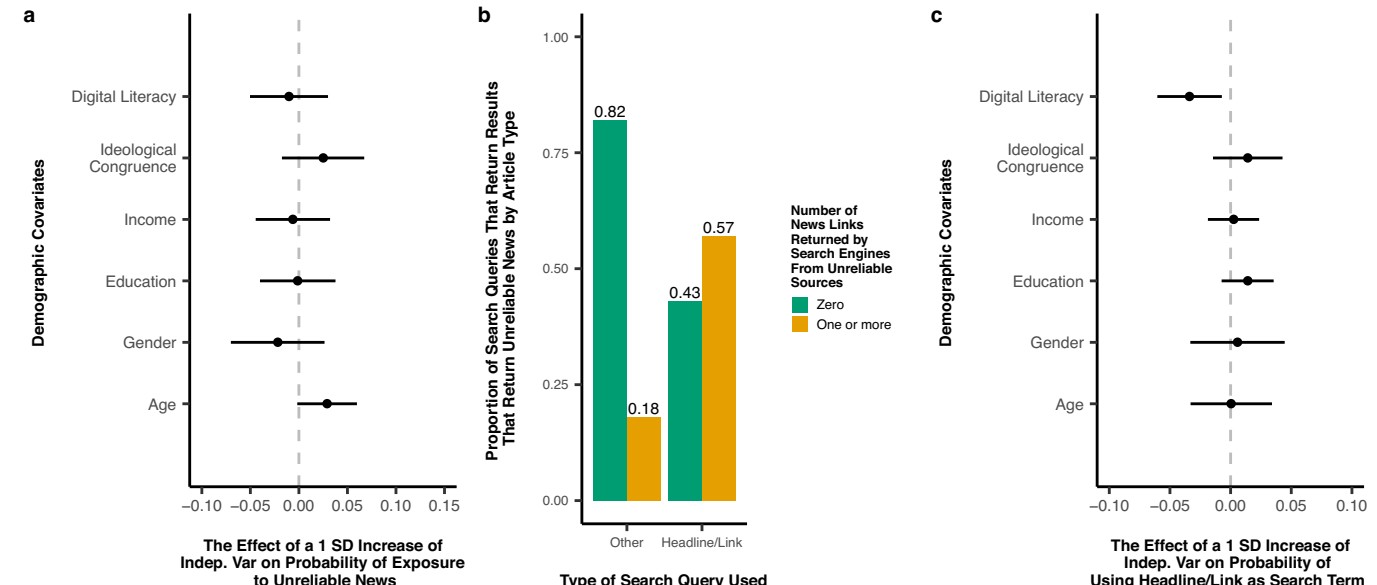

**Extended Data Fig. 5 | Who is exposed to unreliable news sites when evaluating misinformation online? (Study 5) - Excluding original article in search results analysis.** Panel a presents the effect of demographic variables on the probability of exposure to unreliable news sources when searching online about false/misleading news articles and 95 percent confidence intervals during Study 5 (N = 498). Panel b presents the proportion of online searches individuals engage in (N = 930) when searching online about a false/misleading article returns different levels of unreliable news sites by the Google search engine. We present these proportions for those who use the headline of the article or the link of the article and those who use another query. Panel c presents the effect of demographic variables on the probability of using the headline/lede or unique URL when searching online about false/misleading news articles and 95 percent confidence intervals during Study 5 (N = 930). All effects are estimated using ordinary least squares with article fixed effects and standard errors clustered at the individual and article level.

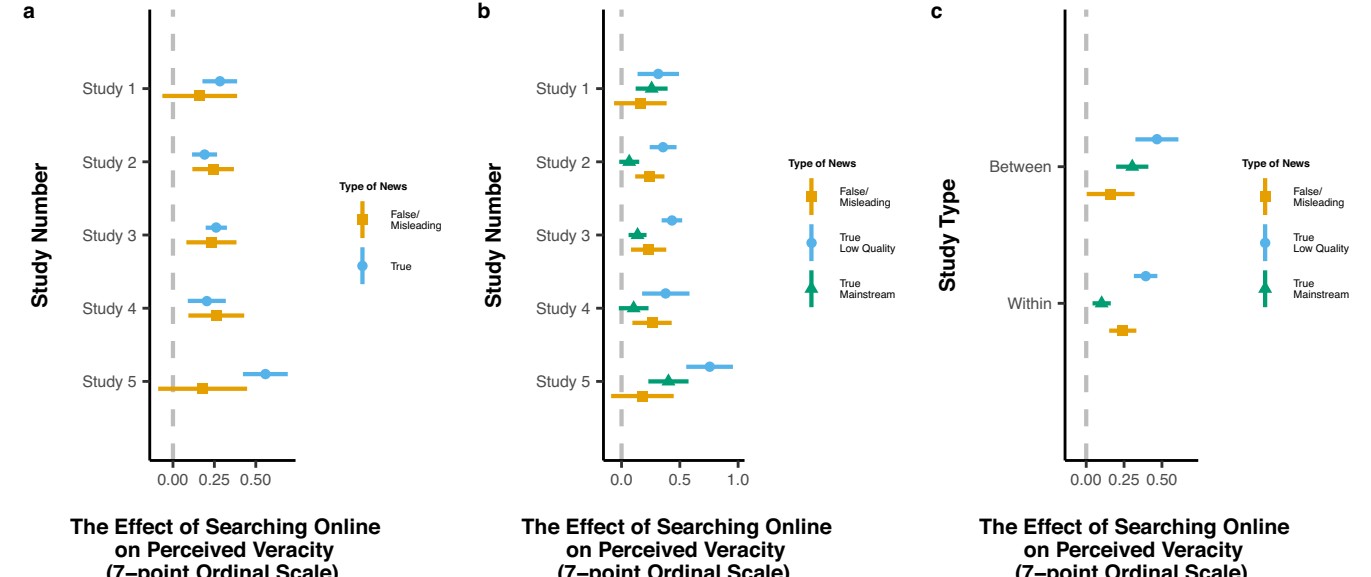

**The Effect of Searching Online on Perceived Veracity (7–point Ordinal Scale)**

**The Effect of Searching Online on Perceived Veracity (7–point Ordinal Scale)**

**The Effect of Searching Online on Perceived Veracity (7–point Ordinal Scale)**

**Extended Data Fig. 6 | Fig. 4 with 7-point ordinal scale.** Panel a presents the effect of rating true news as true and false/misleading news as true and 95 percent confidence intervals using a seven-point ordinal scale during Studies 1 (N = 6,269, N = 2,275), 2 (N = 6,046, N = 2,020), 3 (N = 5,098, N = 1,964), 4 (N = 1,420, N = 772), and 5 (N = 3,141, N = 1,485). Panel b the effect of rating true news as true from low-quality sources, true news as true from mainstream sources, and false/misleading news as true and 95 percent confidence intervals during Studies 1 (N = 2,782, N = 3,487, N = 2,275), 2 (N = 2,596, N = 3,450, N = 2,020), 3 (N = 2,490, N = 3,418, N = 1,964), 4 (N = 516, N = 904, N = 772), and 5 (N = 1,350, N = 1,791, N = 1,485). Panel c presents the effect of rating true news as true from low-quality sources, true news as true from mainstream sources, and false/misleading news as true and 95 percent confidence intervals for between-respondent experiments (Studies 1 and 5) (N = 4,132, N = 5,278, N = 4,756) and within-respondent experiments (Studies 2–4) (N = 5,602, N = 7,702, N = 3,760).

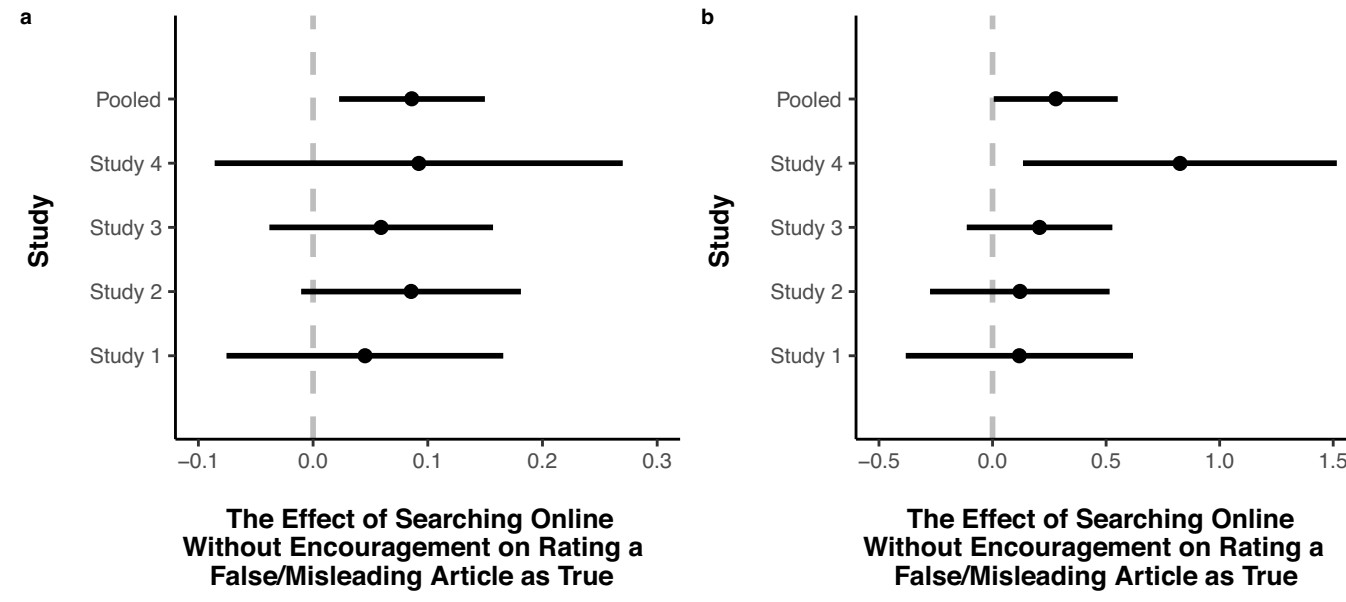

**The Effect of Searching Online Without Encouragement on Rating a False/Misleading Article as True**

**The Effect of Searching Online Without Encouragement on Rating a False/Misleading Article as True**

**Extended Data Fig. 7 | Effect of SOTEN about false/misleading news articles when unprompted.** This figure presents the effect of SOTEN unprompted on rating a false/misleading article as true and 95 percent confidence intervals using a categorical measure in panel a and a 7-point ordinal scale in panel b for Studies 1 (N = 1,145), 2 (N = 1,010), 3 (N = 982), 4 (N = 386, and all four studies pooled together (N = 3,523). When pooled together we observe that searching online increases rating a false/misleading news article by 0.086 (P = 0.0102, Cohen's $D$ = 0.18, $N$ = 3,523) and increases perceived veracity on a seven-point ordinal scale by 0.278 (P = 0.0463, Cohen's $D$ = 0.16, $N$ = 3,523).

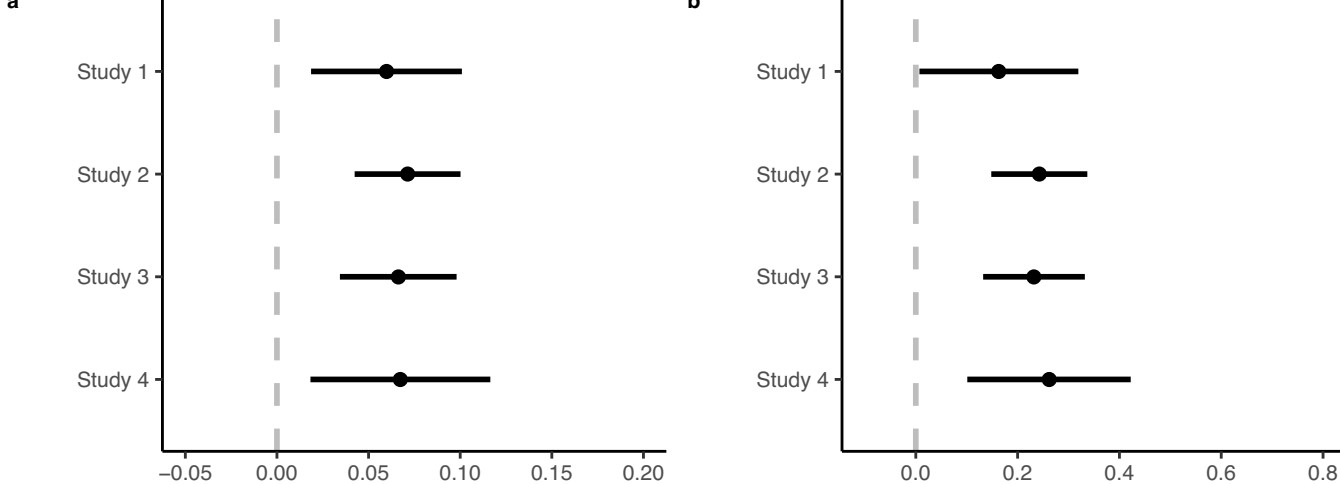

**a**

**Effect of Searching Online on Probability
of Rating Misinformation as True**

**b**

**Effect of Searching Online on the Perceived
Veracity of Misinformation (7–point scale)**

**Extended Data Fig. 8 | The effect of searching online on belief in
misinformation across Study 1 through 4 using the preregistered models.**
Panels a and b present the average treatment effect of SOTEN on rating false/
misleading articles as true and 95 percent confidence intervals during Studies
1 (N = 2,275), 2 (N = 2,020), 3 (N = 1,964), and 4 (N = 772) using our preregistered
model, which only clustered standard errors at the respondent level. All effects
are estimated using ordinary least squares with article fixed effects and
standard errors clustered at the individual level. Panel a presents the effect of
SOTEN on rating misinformation as true and 95 percent confidence intervals
for Study 1 ($P < 0.0001$), 2 ($P < 0.0001$), 3 ($P < 0.0001$), and 4 ($P < 0.0001$).
Panel b presents the effect of SOTEN on a 7-point ordinal scale of veracity
and 95 percent confidence intervals for Study 1 ($P < 0.0001$), 2 ($P < 0.0001$),
3 ($P < 0.0001$), and 4 ($P < 0.0001$).

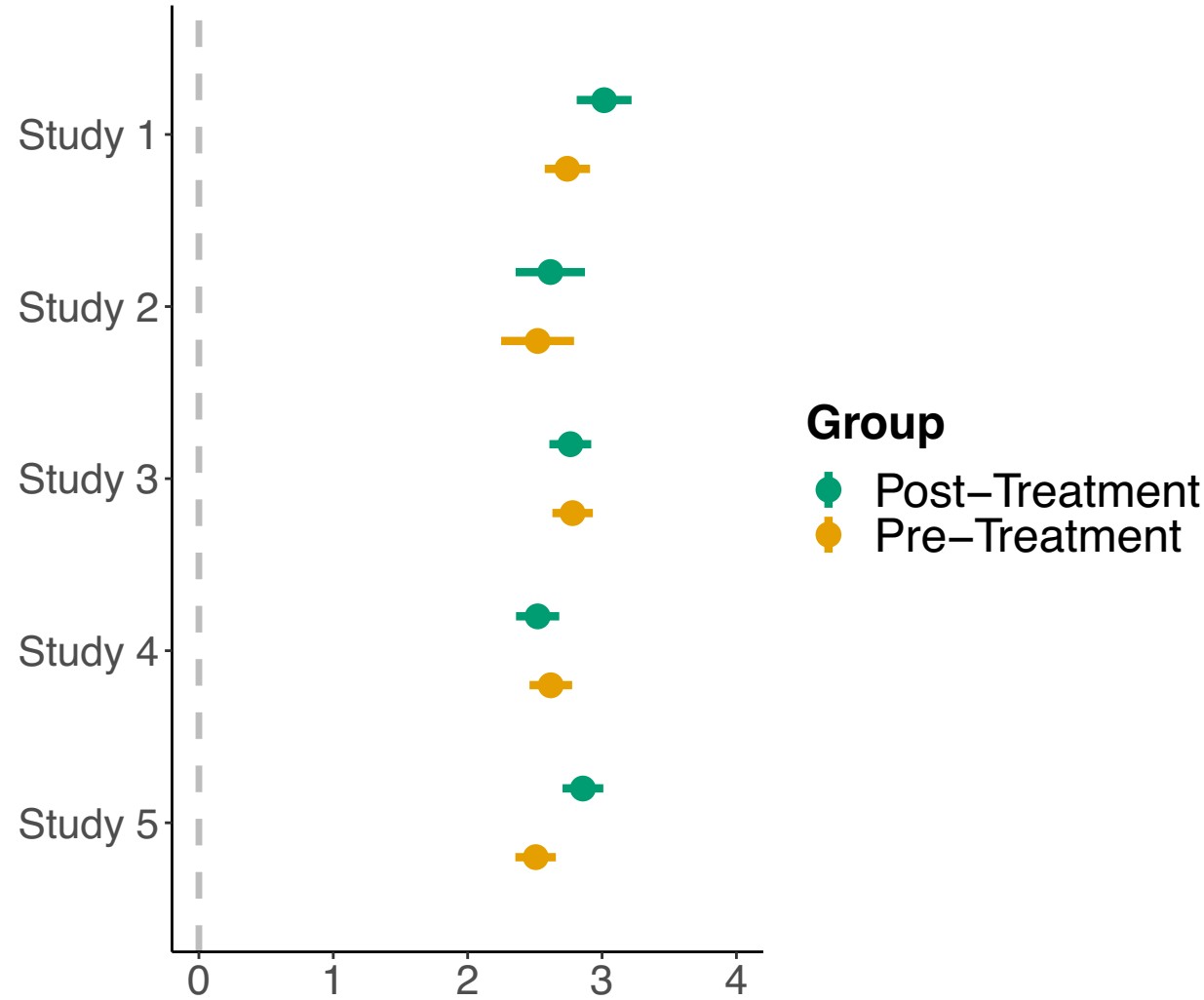

**Effect of Rating Misinformation as True on
Seven–Point Ordinal Scale Veracity Rating**

**Extended Data Fig. 9 | Predicting seven-point ordinal scale with categorical rating for evaluation of false/misleading articles.** We predict the rating of an article on a 7-point scale using our categorical measure using a simple linear regression (ordinary least squares). This figure presents the effect of rating a false/misleading article as true on rating a false/misleading article as true on the 7-point ordinal scale and 95 percent confidence intervals. Pre-treatment, rating a false/misleading article as true increases the 7-point ordinal scale by 2.51 (Study 1; $N = 1,145$; $P < 0.0001$), 2.62 (Study 2; $N = 1,010$; $P < 0.0001$), 2.78 (Study 3; $N = 982$; $P < 0.0001$), 2.52 (Study 4; $N = 386$; $P < 0.0001$), and 2.74 (Study 5; $N = 877$; $P < 0.0001$). Post-treatment, rating a false/misleading article as true increases the 7-point ordinal scale by 2.86 (Study 1; $N = 1,130$; $P < 0.0001$), 2.52 (Study 2; $N = 1,010$; $P < 0.0001$), 2.76 (Study 3; $N = 982$; $P < 0.0001$), 2.61 (Study 4; $N = 386$; $P < 0.0001$), and 3.02 (Study 5; $N = 608$; $P < 0.0001$).

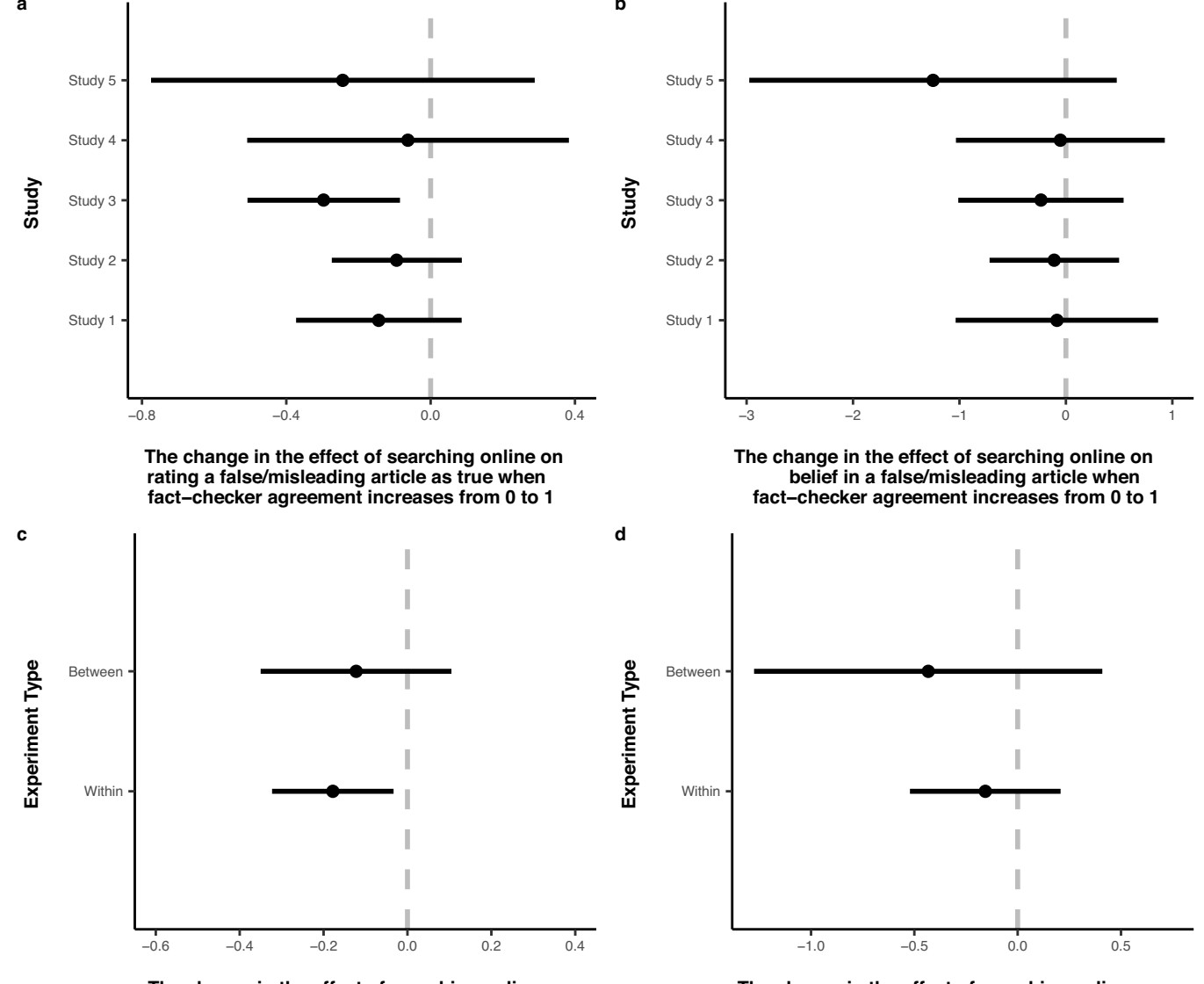

**Extended Data Fig. 10 | Does the effect of SOTEN increase with fact-checker agreement across Studies 1 through 5?.** Panels a and b present the change (with 95 percent confidence intervals) in the effect of searching online on rating a false/misleading article as true when fact-checker agreement increases from 0 to 1 during Studies 1 (N = 2,275), 2 (N = 2,020), 3 (N = 1,964), 4 (N = 772) and 5 (N = 1,485) using a categorical scale (Panel a) and a 7-point ordinal scale (Panel b). Panels c and d present the change (with 95 percent confidence intervals) in the effect of searching online on rating a false/misleading article as true when fact-checker agreement increases from 0 to 1 during between-respondent experiments (Studies 1 and 5; N = 3,760) and within-respondent experiments (Studies 2, 3, and 4; N = 4,756) and 5 (N = 1,485) using a categorical scale (Panel c) and a 7-point ordinal scale (Panel d).

# Reporting Summary

## Statistics

For all statistical analyses, confirm that the following items are present in the figure legend, table legend, main text, or Methods section.

| n/a | Confirmed | |
|---|---|---|
| ☐ | ☒ | The exact sample size ($n$) for each experimental group/condition, given as a discrete number and unit of measurement |
| ☐ | ☒ | A statement on whether measurements were taken from distinct samples or whether the same sample was measured repeatedly |
| ☐ | ☒ | The statistical test(s) used AND whether they are one- or two-sided<br>*Only common tests should be described solely by name; describe more complex techniques in the Methods section.* |
| ☐ | ☒ | A description of all covariates tested |
| ☐ | ☒ | A description of any assumptions or corrections, such as tests of normality and adjustment for multiple comparisons |
| ☐ | ☒ | A full description of the statistical parameters including central tendency (e.g. means) or other basic estimates (e.g. regression coefficient) AND variation (e.g. standard deviation) or associated estimates of uncertainty (e.g. confidence intervals) |
| ☐ | ☒ | For null hypothesis testing, the test statistic (e.g. $F$, $t$, $r$) with confidence intervals, effect sizes, degrees of freedom and $P$ value noted<br>*Give P values as exact values whenever suitable.* |
| ☒ | ☐ | For Bayesian analysis, information on the choice of priors and Markov chain Monte Carlo settings |
| ☐ | ☒ | For hierarchical and complex designs, identification of the appropriate level for tests and full reporting of outcomes |
| ☐ | ☒ | Estimates of effect sizes (e.g. Cohen's $d$, Pearson's $r$), indicating how they were calculated |

*Our web collection on statistics for biologists contains articles on many of the points above.*

## Software and code

Policy information about availability of computer code

| Data collection | Two web extensions were used in the data collection process for Study 5. They are titled: "Search Engine Results Saver" and the "URL Historian." They are available on the chrome webstore for free. No other code or software was used. |
|---|---|
| Data analysis | R (4.2.3) and RStudio (2023.03.0+386) was used to clean and analyze the data. We created our own code to do so. |

For manuscripts utilizing custom algorithms or software that are central to the research but not yet described in published literature, software must be made available to editors and reviewers. We strongly encourage code deposition in a community repository (e.g. GitHub). See the Nature Portfolio guidelines for submitting code & software for further information.

## Data

Policy information about availability of data

All manuscripts must include a data availability statement. This statement should provide the following information, where applicable:
- Accession codes, unique identifiers, or web links for publicly available datasets
- A description of any restrictions on data availability
- For clinical datasets or third party data, please ensure that the statement adheres to our policy

Data and materials for all of the studies are available at https://github.com/SMAPPNYU/Do_Your_Own_Research.

# Research involving human participants, their data, or biological material

Policy information about studies with human participants or human data. See also policy information about sex, gender (identity/presentation), and sexual orientation and race, ethnicity and racism.

| | |
|---|---|
| Reporting on sex and gender | Gender was used as a control variable in some of the analyses conducted in the paper. Gender was determined by self-reporting. We do provide disaggregated gender data in the source data. We do not find any evidence in our studies that the findings only applied to one sex or gender. We did not pre-register any hypotheses regarding gender and we stuck close to our pre-registration. |
| Reporting on race, ethnicity, or other socially relevant groupings | Participants themselves provided demographic information that we controlled for in our study. We controlled for the following variables: age, education, income, political ideology, and gender. The following questions asked individuals for this data:<br><br>Age: What is your age?<br><br>Ecucation: We asked individuals to self-identify their highest degree earned.<br><br>Income: We asked individuals to self-identify their income from last year.<br><br>Political Ideology: Where would you place yourself on this scale? Extremely Conservative - Extremely Liberal<br><br>Gender: What is your gender?<br><br>We ran experimental studies that sampled a representative sample of individuals using quota-sampling and randomized the treatment. We then controlled for these demographic variables to improve the precision of our average treatment effect. |
| Population characteristics | In all of the studies we sampled individuals living in the United States. In the first four of our studies and the sixth study these individuals were recruited by Qualtrics. These samples were representative. We quote-sample respondents based on age, gender, and education. The sample for the fifth study was recruited using Mechanical Turk. This sample was not representative and was not quota-sample based on demographic variables. Balance tables for each study including this demographic information is listed in the methods section of the main text.<br><br>By sampling individuals through online opt-in surveys we do understand that we are oversampling highly online individuals, but this is our target population. We are most interested in frequent users of the internet who are most likely to consume online news. |
| Recruitment | Participants were recruited by Qualtrics, an online survey platform, and Amazon's Mechanical Turk. In both of these cases, participants were told what they would be asked to do in the survey and could opt out at any time. Given these internet surveys use opt-in panels and we know they are the less accurate than probability sampling, we must be cautious when reporting our results. For example, we expect the behavior of our respondents who self-selected into the survey to differ from those drawn with known probability from a well-specified population. Therefore, it is possible and likely this convenience sample is different in possibly unmeasured ways. Therefore, we only report results from analyses with that we can using non-probability sampling.<br><br>Although we should be cautious when making experimental inferences using an opt-in non-probability samples from Qualtrics, previous work has found that about 90% of effects identified using a gold-standard probability sample are similar to effects identified by an opt-in Qualtrics panel.<br><br>A major issue in online opt-in surveys is that the behaviors of those who opt-in to and join multiple panels to earn incentives may put much less effort into tasks at hand and are more likely to guess to save time and maximize their payment. To test if this would affect our main results we ran a parallel survey and paid respondents additional payments for correct answers to our veracity question, but did not find much of any difference in their responses. Therefore, we do not believe that a lack of effort explains the results we find. Recent work has also shown that experimental results from these non-probability samples are often comparable to those found in population samples. Given this previous work, the results we present are not likely to be different if we had used probability-sampling.<br><br>An additional possible issue is that we may have different levels of attrition in the control and treatment groups in a few of our studies. We report dropout levels and balance tables for every study in our paper to provide evidence that we do not believe this to be an issue.<br><br>An added advantage of using online sampling is that it predominately recruits those in whom we are actually most interested: in, frequent users of the internet who are most likely to consume online news. Thus even if our results are less likely to be generalizable to overall population, they are still likely to be generalizable to the population that consumes news online more rather than other recruiting techniques such as in-person surveys. |
| Ethics oversight | Study 1 was approved by NYU IRB protocol IRB-FY2019-3511<br>Study 2 was approved by NYU IRB protocol IRB-FY2019-3511<br>Study 3 was approved by NYU IRB protocol IRB-FY2019-3511<br>Study 4 was approved by NYU IRB protocol IRB-FY2019-3511<br>Study 5 was approved by NYU IRB protocol IRB-FY2021-5608<br>Study 6 was approved by a modified NYU IRB protocol IRB-FY2019-3511<br>We received informed consent from all participants in Studies 1-6. |

Note that full information on the approval of the study protocol must also be provided in the manuscript.

# Field-specific reporting

Please select the one below that is the best fit for your research. If you are not sure, read the appropriate sections before making your selection.

☐ Life sciences  ☒ Behavioural & social sciences  ☐ Ecological, evolutionary & environmental sciences

For a reference copy of the document with all sections, see nature.com/documents/nr-reporting-summary-flat.pdf

# Behavioural & social sciences study design

All studies must disclose on these points even when the disclosure is negative.

| Study description | Quantitative Experimental Studies.<br><br>In Study 1, we tested whether SOTEN affects belief in misinformation in a randomized controlled trial that ran for ten days. During this study, we asked two different groups of respondents to evaluate the same false/misleading or true articles in the same 24-hour window, but only one after searching online.<br><br>Study 2 ran similarly to Study 1, but over 29 days between November 18, 2019 and February 6, 2020. In each survey that was sent in Study 1, we asked respondents in the control group to evaluate the third article they received a second time, but only after looking for evidence online (using the same directions to search online that participants in Study 1 received).<br><br>Study 3 replicated Study 2 using the same materials and procedure, but was run between March 16, 2020 and April 28, 2020, three to five months after the publication of each these articles.<br><br>Study 4 extended Study 2 by asking individuals to evaluate and re-evaluate highly popular misinformation strictly about Covid-19 after searching online. This study was run over eight days between May 28, 2020 to June 22, 2020.<br><br>Study 5 was almost identical to Study 1, but we used a custom plug-in to collect digital trace data and encouraged respondents to specifically search online using Google (our web browser plug-in could only collect search results from a Google search result page). Similar to Study 1, we measured the effect of SOTEN on belief in misinformation in a randomized controlled trial that ran on twelve separate days from July 13, 2021 to November 9, 2021, during which we asked two different groups of respondents to evaluate the same false/misleading or true articles in the same 24-hour window. The treatment group was encouraged to search online, while the control group was not.<br><br>Study 6 tests if the search effects we identify on belief in false/misleading and true articles still hold when we remove the instructions we present to respondents. To this end, we ran an experiment similar to Study 1, but we add two other treatment arms in which we encourage individuals to search online to evaluate news. |
|---|---|
| Research sample | In the first four of our studies our sample of those living in the United States is recruited by Qualtrics. The sample is representative. We quote-sample respondents based on age, gender, and education. The final study was recruited using Mechanical Turk. This sample was not representative and was not quota-sample based on demographic variables.<br><br>By sampling individuals through online opt-in surveys we do understand that we are oversampling highly online individuals, but this is our target population. We are most interested in, frequent users of the internet who most likely to consume online news.<br><br>Balance tables for each study including this demographic information is listed in the methods section of the main text. |
| Sampling strategy | Individuals are randomly sampled. No statistical methods were used to predetermine sample size. Generally the sample sizes used (N > 1000) are large enough to identify small effects (Cohen's D above 0.2) using our models. |
| Data collection | Respondents took these surveys online from either their desktop or mobile phone. The respondents did not interact with the researcher. |
| Timing | The timing of each study can be found below:<br>Study 1: November 21st, 2019 to January 7, 2020<br>Study 2: November 18th, 2019 to February 6th, 2020<br>Study 3: May 28th, 2020 to June 22nd, 2020<br>Study 4: March 16th, 2020 to April 28th, 2020<br>Study 5: July 13th, 2021 to November 9th, 2021<br>Study 6: August 10th, 2022 to September 11th, 2022 |
| Data exclusions | No data was excluded from the analysis. |
| Non-participation | We report varied levels of non-participation in our five studies. These non-participants could have declined to participate in the survey, dropped out after starting the survey, or were dropped because they failed an attention check. Participants who declined to participate or dropped out of the study, did not notify us why they refused to participate. The percentage of non-participation can be found below:<br>Study 1: 82%<br>Study 2: 82% |

Study 3: 74%
Study 4: 76%
Study 5: 75%
Study 6 (only reported in supplementary materials): 78%

Randomization | Individuals were randomly allocated to different experimental groups.

# Reporting for specific materials, systems and methods

We require information from authors about some types of materials, experimental systems and methods used in many studies. Here, indicate whether each material, system or method listed is relevant to your study. If you are not sure if a list item applies to your research, read the appropriate section before selecting a response.

## Materials & experimental systems

| n/a | Involved in the study |
|-----|------------------------|
| ☒ ☐ | Antibodies |
| ☒ ☐ | Eukaryotic cell lines |
| ☒ ☐ | Palaeontology and archaeology |
| ☒ ☐ | Animals and other organisms |
| ☒ ☐ | Clinical data |
| ☒ ☐ | Dual use research of concern |
| ☒ ☐ | Plants |

## Methods

| n/a | Involved in the study |
|-----|------------------------|
| ☒ ☐ | ChIP-seq |
| ☒ ☐ | Flow cytometry |
| ☒ ☐ | MRI-based neuroimaging |

