## [Peer Review File · Nature]

Manuscript Title: Do Your Own Research: How Searching Online to Evaluate Misinformation Can Increase Its Perceived Veracity

Reviewer Comments & Author Rebuttals

Reviewer Reports on the Initial Version:

Referees' comments:

Referee #1 (Remarks to the Author):

The authors investigate whether searching online to assess the accuracy of news articles actually decreases the accuracy of judgments. In a final study with trace data from searches, they find a correlation between inaccurate judgments and bad search results.

In general, I think the authors are probably correct that it's much more important to teach people *how* to search than it is to encourage them *to* search (and I assume that most would agree with this, actually). However, I'm not sure that the authors have produced the right analyses of their data to truly evaluate the policy of encouraging people to search for more information online. In particular, they have restricted their analyses to searching of misinformation and have not produced an analysis of cases where the participants did searches for (apparently) true content.

To me, the central question is not simply whether searching for more information on false content will increase belief, but whether this outweighs the (I assume) positive effect of searching for more information on true content. That is, to evaluate whether the recommendation of "go search things" is a good recommendation, one cannot subset on whether the claims are known to be false or true after the fact. If searching online to evaluate true information increases acceptance of it and this effect is about the same as that which is presented in the manuscript, then the overall benefit of the policy depends on the extent to which people are being exposed to true versus false information. And, given that most people are generally exposed to mostly true stuff, the policy might actually be a net positive despite what the authors report.

So, we need to know what happens with the true content and whether the effect size is larger/smaller/the same as for misinformation. Without that, I don't think we can evaluate the policy recommendation, which appears to be the most significant contribution of this paper.

Another issue, in terms of how this research maps on to the relevant recommendations that are made, is that perhaps the most central element of online search is not being modeled here: That which influences the choices people make about *what* to put effort into searching/investigating. That is, people in the experiments did not have discretion in what they decide to search, which is naturally not true in the world. What the authors are evaluating here is an intervention where people are forced to search for more information on topics – this is not something that I have seen recommended. This isn't to say that the present project isn't interesting or worthwhile; rather, my point is simply that it's hard to evaluate the stated policy based on these experiments. Something the authors could speak to.

Given the foregoing, suitability for Nature perhaps rests more on what this teaches us about the underlying psychological processes. Unfortunately, I don't think it reaches the bar for that. (the underlying psychology being that people sometimes believe things that they read)

This is essentially a historical snapshot of the ability of Google search algorithms to produce high quality information. [The authors acknowledge this to some extent, e.g., "A study run months or

years after publication would test the impact of a different set of search engine results, and it would be impossible to replicate the original results present in the period of most likely exposure.”] So, in a way, the investigation is fundamentally limited by caveats about what sort of things people are searching for and the algorithm itself. For example, the same experiment could be run in 5 years and produce quite different results because the search algorithm has been modified. Or, perhaps, misinformation that comes in different forms produce different results. Etc.

What this means, ultimately, is that this study is an existence proof for the potential backfire of online searches. I do think that this is a really interesting and important finding, but I could imagine the nuance of this being missed by people if it isn't made very clear upfront. [I should note, though, that I thought the authors did a really commendable job of producing a well balanced manuscript overall.]

Additional comments

- “Building off of this, media literacy programs should replace a general focus on online search with more targeted techniques that teach individuals how to use proper search terms and identify quality news sources” Do any actually claim this? Suggestions about searching online (e.g., footnotes 3 and 5) tend to qualify the recommendation by saying that one should look to reputable/credible sources. So it's not that people are claiming that search itself is a good idea.

- I think that the use of percent increase in the abstract might be a bit inflationary. The overall effects are actually quite small and p-values consistent hover around $p = .05$. The authors may want to also report some Bayesian analyses to get a sense of the amount of evidence in these studies, because it doesn't seem all that strong to me. Another option would be to report some standardized effect sizes (e.g., Cohen's D).

- Also, and this is not accusatory, but it's very important that there aren't other relevant studies in the file drawer given how “close” several of these results are to being non-significant. I do commend the authors for preregistering their studies (and, in fact, for using an analysis that is more conservative than what was preregistered!). I also really liked the principled headline selection process. These were very rigorous experiments and I was really impressed by what the authors put together here. My general point is that I would probably conclude from the paper that searching online for misinformation *can* increase belief in falsehoods. It probably doesn't *always* do so, and the effect isn't very big, but this is still an important point to make. I would not be surprised if other studies with different specifications produced different results.

- “We find that the direction of our results do not change when using the false/misleading articles with a robust mode.” I suppose, but it should be noted here that the effect is not significant in 2 of 4 for probability of rating true and 1 of 4 for continuous measure.

- The authors said the following in the instructions: “If you cannot find evidence about the claim from a source that you trust, you should try to find the most relevant evidence about the claim you can find from any source, even one you don't trust.” This seems problematic to me. Particularly since they note in the main text: “In the survey, respondents are asked to find credible sources to help them evaluate the veracity of the news articles, mirroring many media literacy interventions. Why then do individuals allow low-quality sources to influence their evaluation?” Sure, participants were asked to find credible sources... but they were also asked to just find evidence regardless of quality (assuming they couldn't find credible sources – which seemed to be true much of the time).

- Just speculating on the mechanism, it seems that search may raise confidence that the individual knows the answer. However, this is false confidence when the reality is difficult to discern. This is consistent with what the authors found: “Among those who could not determine the veracity of the article initially, more individuals incorrectly changed their evaluation to true than to

false/misleading after being required to search online." It also makes sense that the effect seems to be greater for headlines where there is more ambiguity about what is true v false (according to fact-checkers).

- Was there differential dropout in treatment versus control?

- "By asking respondents in both the control and treatment group to install a custom web extension that collected their web browsing behavior, we were able to collect digital trace data associated with 73% of evaluations of false/misleading articles in the treatment group and 91% of evaluations of false/misleading articles in the control group" What's the source of the difference here?

- "Only 12% of search queries about true articles return at least one unreliable news link among the top ten results, whereas 32% of search queries about false/misleading articles return at least one unreliable news link among the top ten results." This reinforces the need for a comparison between true and false articles.

- "Searching online also increased the average score by 0.15 ($F=12.149$, $P=0.0467$) on a 4-point ordinal scale and 0.16 ($F=11.430$, $P=0.2155$) on a 7-point ordinal scale" The latter isn't an increase but is stated here as if it is.

I always sign my reviews (unless the authors are blinded),
Gordon Pennycook

Referee #2 (Remarks to the Author):

Generally, this is a methodologically sound study investigating an important topic, with a surprising and actionable result. Given the current focus on mis/disinformation and media literacy, it's extremely timely and significant. As I understand it, the researchers found:

- Searching online to evaluate misinformation (SOEM) increased the likelihood that a study participant would identify a false story as true, in one case, as much as 26%

- Findings were robust across a number of methodologies (time delay between initial exposure and re-evaluation), and effect size increased in the most involved methodology, which forced users to perform SOEM

- A possible mechanism exists through which participants search for information in a way likely to turn up low quality information specifically when they search for information about misinformation stories

- This suggests a refinement in media literacy training where users are urged to perform SOEM without including an article's title or URL, so they can find evaluation of the claims made instead. This also might suggest a path forward for search engines in cases where they are highly likely to return low quality information.

Assuming I have the basics of the paper correct, some thoughts about room for improving an already very strong paper:

- Authors should characterize the stories in the text as highly popular stories, given that they were the top stories on a given day, shared from various US news sources. Obviously there are differences between low quality and high quality sources in terms of their reach, but the popularity of these stories is important to note. There's a significant difference in implications of this work if we're encountering obscure news versus widely disseminated news. Also adds the complicated reminder that popular news from US news sources routinely contains mis/disinformation.

- Authors chose to evaluate the effects of SOEM on misinformation stories. What are the effects of SOEM on true stories? Are there cases where users try to verify true stories and end up persuading themselves that they are false? While not the focus of this study, it seems important to share any insights on this topic, as one policy implication of this study is that we should be far more cautious in urging people to "do their own research". If there are significant cases where people turn away from true information (as some of the evaluation of studies 1-4 seems to suggest), that emphasizes this policy point.

- Centering Golebiewski and boyd's notion of data voids is helpful, but it might be worth exploring their argument in more depth and considering some key implications. Golebiewski and boyd believe that right wing actors are often working to seize on data voids - they give the example of the shooting in Sutherland Springs, TX and identify situations where white supremacists created content very rapidly to pull viewers from the data void.

I think what authors may be seeing is an ecosystem of mis/disinfo similar to that documented by Benkler et al in Networked Propaganda. When users are finding low quality information by searching for the title or URL of an article they're trying to evaluate, it's likely they are stepping into traps consciously set by misinformation brokers. Misinfo pages often link to one another to increase their search engine rankings and their perceived credibility. Searching for a title or URL, rather than for an article's claims, may be a particularly dangerous strategy when these misinfo ecosystems have been set up to trap the unwary.

How common was this search strategy? I.e., what percent of users used URLs in their searches?

- Value of Fleiss's Kappa doesn't indicate a huge level of agreement between professional factcheckers - the authors note that this level of agreement is higher than in similar studies. Might be worth noting that it's surprisingly challenging to validate content, even by professionals, in real time. This also helps explain why it's so hard for Google and other search engines to flag misinfo, especially rapidly moving misinfo, as it's hard to get agreement that something is, in fact, misinfo.

- It might be helpful to cite:

- Derakshan and Wardle on the definition of mis/dis/malinfo

- Newly published Marwick on QAnon

(<https://journals.sagepub.com/doi/full/10.1177/14614448221090201>) - it's more recent and more thorough than the paper you cite

- More qualitative researchers on hypothesis that fact checking can cause harms - Francesca Tripodi and Barbara Fister both are good on this topic.

- It seems like there's a great deal of data that didn't make it into the paper - asking users their search strategies, keywords used, etc. as well as behavioral data from study 5. Would the paper benefit from using at least some concrete examples of searches that generated higher and lower quality results? Suspect the team in planning more detailed analysis going forward - if not, releasing this data to allow more detailed analysis by collaborators would be incredibly helpful.

Excellent paper - important to have this out in the world. Thanks for the chance to review.

Ethan Zuckerman, UMass Amherst, ethanz@umass.edu

Referee #3 (Remarks to the Author):

Summary of the key results: This paper focuses on the impact of search on assessments of accuracy for articles containing misinformation. The authors start from two important observations 1) a disproportionate amount of attention has been paid to social media while considering information channels that aide the spread of misinformation and 2) media literacy advice commonly suggests that readers should do their research before believing whether a piece of information is correct. Both are excellent observations and provide sufficient motivation for the

work. (though as I will mention briefly later, I think #2 is not really fully addressed).

They carry out an impressive number of studies to show that searching can counterintuitively lead to less reliable assessments. their fifth study provides intuition as to why this is the case. as they point out, "that the probability an individual believes misinformation is substantially higher than the control group among respondents who are exposed to at least one unreliable news site, but it is not higher among those in the treatment group who are only exposed to very reliable news sites." So individuals are using evidence from unreliable sources to find support for articles that spread misinformation.

The studies span a broad range: within and between subjects, new vs old misinformation, as well as misinformation in a particular domain area. their final study links web trace data with the experimental data to delve into potential explanations for the finding. here they focus on motivated reasoning and media literacy. I believe this section of the paper has some holes and needs the most work (more on this later).

Originality and significance: if not novel, please include reference

The paper is novel. While there have been some interesting auditing studies focused on search engines and the role that they play in our information ecosystem, as far as I can tell, this is the first study to perform such an audit in the context of fact checking by ordinary individuals.

Data & methodology: validity of approach, quality of data, quality of presentation

I would like to applaud the authors for performing various studies to investigate this important question. The consistency of findings observed across the studies is encouraging. However study 5 requires more work, both in terms of better articulating how the two analyses address motivated reasoning and media literacy and in terms of acknowledging alternative explanations. A bit more on this below:

- If I understand the write up correctly, the authors claim that both exposure of misinfo (bad results showing up in google results) and belief in it (making the wrong assessment after seeing it in the results) can indicate motivated reasoning (in so far as there is a connection to ideology). I can see the case for the latter but I need the authors to better argue why the former is connected to motivated reasoning. Getting exposed to search results do not say anything (at least directly) about one's processing of info. Overall, a significant rewrite of this section is needed to improve its presentation. My concerns here also relate to acknowledging alternative explanations. Indeed which search results one sees for a query can be driven by personalization but a huge component of that personalization is location. The authors do not collect this data or consider it in their theoretical models. One might be served results based on their location and given the ideological geographical segregation, the factors that dictate google results might be less about the person themselves and more about where they live. There also seems to me that there is a simpler explanation of why we see worse results for people who see low credibility domains in their results. As you noted in your codebook, people are instructed to use info even if it is coming from a domain they don't trust. As you also noted, low credibility publishers form a network and reshare each others' content. So the closest text might be coming from those sites (even if one is not searching for the url only). Given your directions to the subjects, isn't this finding expected? This is related to media literacy dimension but I find it hard to separate this from the quality of the direction provided for the human subjects.

The authors do a fairly good job in presenting their ideas in a clear and concise manner. But there are some improvements needed. These are fairly simple and will help the authors better convince their readers of the claims they are making. A few ideas:

-- The authors provide a clear description of the 5 studies at the end but not in the beginning. I would encourage the authors to move the description: "(1) between-respondent and within-respondent experiments (2) using general misinformation and misinformation about a salient event ..." to the intro and instead have the shorter summary currently provided in the intro to the conclusion. I found myself wondering what those 5 studies were and why they were needed. This

short description would have addressed that early on.

-- A similar idea applies to their characterization of the misinformation assessment tasks. Given their questions and claims, it is really important to provide more information about the guidance provided to the subjects when they are making their assessments. The authors' claims are tightly connected to the effectiveness of media literacy advice but this advice is rarely as simple as "do your research". It generally comes with cues to look for, definitions of credibility etc. So without information on that in the main paper, a skeptical reader might assume the authors picked a straw man version of media literacy advice and question the findings. I think there are important improvements in those codebooks that could have changed at least the magnitude of the effect sizes but this concern is not completely unaddressed by the authors. So my comment here is three-fold: 1) simply provide more information about what you have done in the intro/summary of the experiments, 2) connect that to past research and guidance and discuss how your codebooks are aligned (or not), and 3) discuss in the discussion how changes to the codebook can change some of the findings. For instance, the codebook a) doesn't provide much guidance on what sources should be trusted and b) also explicitly tells the subject to use information from sources they don't trust. These might have shaped people's assessments.

Appropriate use of statistics and treatment of uncertainties

- The authors rely on fact checkers to find a subset of articles that are deemed misinfo. But as they also state in their paper, the interrater reliability is only 0.42, much lower than what is acceptable in content analysis research. The authors then collapse the labels provided by the fact checkers to a single label. One could have instead used other measures such as increase in agreement with a random fact checker (this would mean that you don't flatten but inspect agreement with all). I would like to see the authors consider these other alternatives and discuss the limitations/implications given the disagreement observed even among the experts. There are some nice findings briefly mentioned already (how the direction of results don't change if we consider a subset of articles with stronger agreement) but expert disagreement can be considered an indicator of difficulty and the treatment effects might vary wrt it.

Conclusions: robustness, validity, reliability

- The authors make sweeping statements in their conclusions, in particular about media literacy campaigns. However, the guidance they provide the subjects is rather limited and various studies in content analysis would indicate that codebooks and training provided to the subjects can strongly shape their assessments and accuracy. I think that the authors can qualify their statements while not losing their power.

Suggested improvements: experiments, data for possible revision

- Suggested improvements are listed above and are generally about increased clarity in writing and being clearer about the limitations of the work in the discussion section.
- While the authors are primarily interested in how people assess misinformation, I think it is equally important how they assess correct information. For instance, "do your research" frames might get people to start questioning reliable information. I would like to see analysis of how the treatment affects true articles. Not all studies have this but some do.

References: appropriate credit to previous work?

- Appropriate. Some of the concerns I listed (and the accompanying suggestions) might lead to additional references to media literacy research.

Clarity and context: lucidity of abstract/summary, appropriateness of abstract, introduction and conclusions

- Some important missing details leave the reader hanging (noted above) and the description of study 5 needs to be improved. Other than that, the paper was easy and a pleasure to read.

Overall, despite the concerns I listed, I truly enjoyed reading this paper and think it has various strong elements. I believe most of my concerns can be addressed through a rewrite.

Author Rebuttals to Initial Comments:

To the Editors of *Nature*,

Please find attached the revised manuscript “Do Your Own Research: How Searching Online to Evaluate Misinformation Can Increase Its Perceived Veracity” for consideration for publication in *Nature*.

We thank the editor for the opportunity to revise our paper along the lines suggested, and we hope you share our assessment that we took the concerns of the reviewers seriously and that the revised version is now much improved as a result. All the revised text is in red in the newly revised manuscript. In the decision letter, the editor highlighted that three major revisions should be made: (1) Address the effect of online search on belief in true news (in addition to belief in false/misleading news); (2) Run additional analyses to demonstrate that the main results are not biased by the instructions we give respondents and that these results are likely to generalize to people conducting searches on their own; (3) Report additional analyses regarding the connection of exposure to low quality information and motivated reasoning. In response to these three suggestions from the editor and the reviewers, we have made three major changes to the manuscript:

(1) In the fifth section of the paper (titled “The Impact of Search on Belief in True News” on pages 19-23), we report pre-registered analyses on the effect of searching online on belief in true news in Studies 1-5. We find that while the search effect on belief in true news is similar to false/misleading news in the between respondent studies (1 and 5), the effect on true news is either smaller than on false/misleading news or non-existent across the within respondent studies (2-4). Building off the literature on the search engine information environment, we run new exploratory analyses to investigate whether the search effect on true news has heterogeneous effects depending on the quality of the source of the new story. More specifically, we subset our measurement of the search effect on true articles from mainstream (more reputable) and low-quality (less reputable) sources. We find that the effect of online search on true news is much larger if the article is published by a low-quality source than if published by a mainstream source. In fact, the effect of searching for additional information online about a true mainstream news story is either small or non-existent. We incorporate these results into the final section of the paper. We think these heterogeneous effects paint a comprehensive and complex picture of the online search effect that, taken together, provides novel evidence with which to inform the development of digital media literacy strategies.

Please note that, to reflect our new analysis of true news, we changed “searching online to evaluate misinformation (SOEM)” to “searching to online to evaluate news (SOTEN)” throughout the paper.

(2) To determine whether our results were biased by the instructions we provided respondents, we ran a new, sixth study in August and September of 2022 after having received the reviewers' requests to test three different types of instructions to search for more information online. More specifically, these instructions varied the level of detail regarding the search process. We found similar effects for each type of instruction. We discuss this new study in a footnote on page 4 of the manuscript and provide an expanded explanation of this study and the results in Section R of the Supplementary Materials (pages 95-99 of Supplementary Materials). In addition, we ran analyses that provide suggestive evidence of the search effect when individuals search unprompted. In most of our original studies, we asked individuals in the control group whether they searched online to evaluate information. Using these data, we found that people who searched for more information without prompting were also more likely to rate false/misleading news as true than those who did not search online. We added text discussing this analysis on page 25.

(3) We agree with the reviewer that we cannot isolate motivated reasoning as the mechanism to explain exposure to low-quality information, as our design cannot determine whether or not respondents were actually motivated to seek out ideologically aligned results. To reflect this, we have replaced the term "motivated reasoning" with "ideological congruence" throughout the manuscript. We run analyses that demonstrate ideological congruence is associated with exposure to unreliable news sites (pages 16-17); however, as we discuss on these pages, we cannot adjudicate whether this association is the result of search strategies, motivated reasoning, or search engine personalization. To this end, we think "ideological congruence" is a more accurate description of our line of inquiry.

In addition to these three major revisions, we also addressed each individual reviewer comment. We detail these changes below. We thank the editor and reviewers for their generous and detailed feedback — we hope you and they will share our assessment that our responses to their suggestions added clarity and depth to our manuscript.

Reviewer 1

(1) In general, I think the authors are probably correct that it's much more important to teach people *how* to search than it is to encourage them *to* search (and I assume that most would agree with this, actually). However, I'm not sure that the authors have produced the right analyses of their data to truly evaluate the policy of encouraging people to search for more information online. In particular, they have restricted their analyses to searching of misinformation and have not produced an analysis of cases where the participants did searches for (apparently) true content. To me, the central question is not

simply whether searching for more information on false content will increase belief, but whether this outweighs the (I assume) positive effect of searching for more information on true content. That is, to evaluate whether the recommendation of “go search things” is a good recommendation, one cannot subset on whether the claims are known to be false or true after the fact. If searching online to evaluate true information increases acceptance of it and this effect is about the same as that which is presented in the manuscript, then the overall benefit of the policy depends on the extent to which people are being exposed to true versus false information. And, given that most people are generally exposed to mostly true stuff, the policy might actually be a net positive despite what the authors report. So, we need to know what happens with the true content and whether the effect size is larger/smaller/the same as for misinformation. Without that, I don’t think we can evaluate the policy recommendation, which appears to be the most significant contribution of this paper.

Based on the response of multiple reviewers and editors, we agreed that the manuscript would be greatly strengthened by including the effect of search on evaluating true news (something we originally thought would distract from this paper’s focus on belief in false news) and have accordingly updated the manuscript to report the effect of searching online to evaluate news on belief in true news stories as well.

As described above, in the fifth section of the paper (titled “The Impact of Search on Belief in True News” on pages 19-23), we report pre-registered analyses on the effect of searching online on belief in true news in Studies 1-5. We find that while the search effect on belief in true news is similar to false/misleading news in the between respondent studies (1 and 5), the effect on true news is either smaller than on false/misleading news or non-existent across the within respondent studies (2-4). Building off the literature on the search engine information environment, we run new exploratory analyses to investigate whether the search effect on true news has heterogeneous effects depending on the quality of the source of the new story. More specifically, we subset our measurement of the search effect on true articles from mainstream (more reputable) and low-quality (less reputable) sources. We find that the effect of online search on true news is much larger if the article is published by a low-quality source than if published by a mainstream source. In fact, the effect of searching for additional information online about a true mainstream news story is either small or non-existent. We incorporate these results into the final section of the paper. We think these heterogeneous effects paint a comprehensive and complex picture of the online search effect that, taken together, provides novel evidence with which to inform the development of digital media literacy strategies.

(2) Another issue, in terms of how this research maps on to the relevant recommendations that are made, is that perhaps the most central element of online search is not being

modeled here: That which influences the choices people make about *what* to put effort into searching/investigating. That is, people in the experiments did not have discretion in what they decide to search, which is naturally not true in the world. What the authors are evaluating here is an intervention where people are forced to search for more information on topics – this is not something that I have seen recommended. This isn't to say that the present project isn't interesting or worthwhile; rather, my point is simply that it's hard to evaluate the stated policy based on these experiments. Something the authors could speak to. Given the foregoing, suitability for Nature perhaps rests more on what this teaches us about the underlying psychological processes. Unfortunately, I don't think it reaches the bar for that. (the underlying psychology being that people sometimes believe things that they read)

We agree that the factors impacting what people decide to search (and how) are not modeled in our paper, and that this element is indeed important to study. While we have no measurement of what people decide to search when not in our survey, our paper demonstrates the likely effect of expanding online search as a strategy to combat misinformation, as a number of digital media literacy guides have proposed. Given the lack of empirical evidence with which to evaluate the search effect, we see this study as a key first step in understanding how search engines impact news discernment in real time.

We thank the reviewer for this important comment, and on pages 24-25 we make clear the limitations and possible extensions of our study.

Additionally, in a first attempt to address the question raised by the reviewer, we add an exploratory analysis using data from our control group to provide suggestive evidence as to whether we might expect the search effect to hold without being forced to search. In our control group, we asked individuals whether they, without prompting from the survey, searched online to evaluate information. Paralleling the search effect from the treatment group, we find a similar search effect – people searching for information about false stories are more likely to believe that they are true – from respondents in the treatment group who reported using an online search engine in evaluating the news article relative to those who did not.

Both of these additions can be found on pages 24-25:

“To be clear, there is a related dynamic that is worthwhile to study, but is not fully captured in this design: namely, online users have full discretion, often without encouragement, around which stories or topics to evaluate through online search. While this process should be the subject of future research that builds on what we have learned here, it is the case that our current study captures the impact of the intended effect of numerous digital media literacy guides. More specifically, the digital media literacy guides previously cited aim to expand the use of online

search engines to evaluate the veracity of news, with the explicit goal of reducing belief in misinformation. However, the impact of search has yet to be established, and so while our design does not perfectly capture the effect of disseminating this recommendation “in the wild,” our results indicate the likely effect of the intervention if it were adopted. Ultimately, we measure how beliefs in false/misleading and true articles change when respondents search online compared to when they do not. This allows us to properly measure the effect searching online has on belief in false/misleading and true news stories. While our pre-registered analysis focuses on the treatment groups who were encouraged to search, we also have exploratory analysis using control group data that more closely speaks to the search effect when people have full discretion over what to search. Using these data, we find a similar effect: people who, without encouragement, searched to evaluate misinformation were more likely to believe it (see Section V in the Supplementary Materials). Future work could consider using observational data to measure the behavioral impact of disseminating digital media literacy guides, but we think that a better understanding of the impact of SOTEN is a key first step.”

(3) This is essentially a historical snapshot of the ability of Google search algorithms to produce high quality information. [The authors acknowledge this to some extent, e.g., “A study run months or years after publication would test the impact of a different set of search engine results, and it would be impossible to replicate the original results present in the period of most likely exposure.”] So, in a way, the investigation is fundamentally limited by caveats about what sort of things people are searching for and the algorithm itself. For example, the same experiment could be run in 5 years and produce quite different results because the search algorithm has been modified. Or, perhaps, misinformation that comes in different forms produce different results. Etc. What this means, ultimately, is that this study is an existence proof for the potential backfire of online searches. I do think that this is a really interesting and important finding, but I could imagine the nuance of this being missed by people if it isn’t made very clear upfront. [I should note, though, that I thought the authors did a really commendable job of producing a well balanced manuscript overall.]

We agree that this is indeed the case and that we should be more clear about this. On page 4, we added a footnote addressing this point: “Given this, we are testing this search effect in the time period in which our studies run (from Study 1 in late 2019 to Study 5 in late 2021). It is possible that over time the online information environment may change as the result of new search strategies and/or search algorithms; however, these five studies present remarkably consistent evidence of this effect from 2019 to 2021.”

(4) “Building off of this, media literacy programs should replace a general focus on online search with more targeted techniques that teach individuals how to use proper search terms and identify quality news sources” Do any actually claim this? Suggestions about searching online (e.g., footnotes 3 and 5) tend to qualify the recommendation by saying that one should look to reputable/credible sources. So it’s not that people are claiming that search itself is a good idea.

We agree that these programs ask individuals to find information from credible sources; importantly, our instructions also specified the use of credible sources. We thank the reviewer for pointing out this inconsistency and as a result, have removed the second half of the sentence and instead emphasized that digital media literacy interventions should also focus on recommending how individuals can search for information. Now the last two sentences of the discussion section read:

“Our findings highlight the need for media literacy programs to ground their recommendations in empirically tested interventions, as well as search engines to invest in solutions to the challenges identified here. For example, recent developments in the space — such as the expansion of teaching lateral reading strategies (Breakstone et al., 2021) and Google’s warning when no credible information is available for given search queries — are interesting steps in this direction and deserve further testing.”

(5) I think that the use of percent increase in the abstract might be a bit inflationary. The overall effects are actually quite small and p-values consistent hover around p .05. The authors may want to also report some Bayesian analyses to get a sense of the amount of evidence in these studies, because it doesn’t seem all that strong to me. Another option would be to report some standardized effect sizes (e.g., Cohen’s D).

We thank the reviewer for this comment. We agree that this may be inflationary and so when reporting the effect of online search on belief in false/misleading news or true news, we deleted all uses of percentage increases throughout the paper, as well as added Cohen’s D statistics. The Cohen’s D statistic also replaced the F-statistic.

(6) “We find that the direction of our results do not change when using the false/misleading articles with a robust mode.” I suppose, but it should be noted here that the effect is not significant in 2 of 4 for probability of rating true and 1 of 4 for continuous measure.

We agree that we should be clearer about the significance. The text now reads as follows: “the effect is no longer statistically significant for 2 of the 4 studies using the categorical measure and and 1 of the 4 studies using the continuous measure.”

(7) The authors said the following in the instructions: “If you cannot find evidence about the claim from a source that you trust, you should try to find the most relevant evidence about the claim you can find from any source, even one you don't trust.” This seems problematic to me. Particularly since they note in the main text: “In the survey, respondents are asked to find credible sources to help them evaluate the veracity of the news articles, mirroring many media literacy interventions. Why then do individuals allow low-quality sources to influence their evaluation?” Sure, participants were asked to find credible sources... but they were also asked to just find evidence regardless of quality (assuming they couldn't find credible sources – which seemed to be true much of the time).

We thank the reviewer for this comment and their close reading of our survey instrument. To determine whether our results were biased by the instructions we provided respondents, we ran a sixth separate study in August and September of 2022 to test three different types of instructions to search for more information online. More specifically, these instructions varied the level of detail regarding the search process. We found similar effects for each type of instruction. We discuss this new study in a footnote on page 4 of the manuscript and provide an expanded explanation of this study and the results in Section R of the Supplementary Materials (pages 95-99 of Supplementary Materials). We appreciate the reviewer's comment, as we believe the additional study it prompted contributes to the robustness of our results.

In addition, we deleted the following sentence from the article in order to more accurately reflect the complete instructions respondents were provided: “In the survey, respondents are asked to find “credible” sources to help them evaluate the veracity of the news articles, mirroring many media literacy interventions.”

(8) Just speculating on the mechanism, it seems that search may raise confidence that the individual knows the answer. However, this is false confidence when the reality is difficult to discern. This is consistent with what the authors found: “Among those who could not determine the veracity of the article initially, more individuals incorrectly changed their evaluation to true than to false/misleading after being required to search online.” It also makes sense that the effect seems to be greater for headlines where there is more ambiguity about what is true v false (according to fact-checkers).

This is a good point as to the mechanism. On page 8, we add: “This suggests that searching online may falsely raise the confidence of individuals evaluating false/misleading news.” In response to the last sentence of the comment, we add a new analysis to determine if the search effect changes as the agreement of fact-checkers changes. We ran an interaction model and present the results in Section U of the Supplementary Materials. We find that the search effect does appear to weaken for articles that fact-checkers are most likely to agree are false/misleading. We add a discussion about this to page 31 of the main text:

“To determine if the search effect changes with the rate of agreement of fact-checkers, we ran an interaction model and present that results in Section U of the Supplementary Materials. We find that the search effect does appear to weaken for articles that fact-checkers most agree are false/misleading. Put another way, the search effect is strongest for articles in which there is less fact-checker agreement that the article is false, suggesting that online search may be especially ineffective when the veracity of articles is most difficult to ascertain.”

(9) Was there differential dropout in treatment versus control?

We thank the reviewer for this question. Yes, there was a slight differential in dropout in Study 1 and a much greater difference in Study 5. In the other three studies, everyone was in the same group because it was a within-respondent experiment. In Study 1, 83.2% of those who entered the survey and were in the control group dropped out of the study, while 85.8% of those who entered the survey and were in the treatment group dropped out of the study. The majority of respondents dropped out of the survey at the beginning. Roughly 66% of all respondents who entered the survey refused to consent or did not move past the first two consent questions. Of all respondents who moved past the consent questions, 51% of respondents dropped out of the survey in the control group and 58% of the respondents dropped out of the survey in the treatment group. Roughly 11% of those who did not complete the survey were removed because they failed the attention checks. Taken together, while there was a large dropout rate in general (likely due to the length of the survey relative to typical Qualtrics surveys), the differential dropout between treatment and control was minimal.

In Study 5, only 20% of those in the control group who consented to participate in the survey dropped out of the study, while 62% of those who entered the survey and were in the treatment group dropped out of the study. There was likely a large difference in the Study 5 because respondents had to wait at least five seconds for the web extension that was installed to collect their Google search engine results, likely leading to a much higher dropout rate in the treatment group. This differential attrition does not result in any substantively meaningful differences between those who completed the survey in the treatment and control group, as shown in the methods section. We also do not believe this attrition rate to have an effect on our results, as our within-respondent experiment reports consistently similar results to our between-respondent experiment.

We added a discussion of the dropout rates for Study 1 on pages 34-35 and the dropout rates for Study 5 on pages 39-40.

(10) “By asking respondents in both the control and treatment group to install a custom web extension that collected their web browsing behavior, we were able to collect digital trace data associated with 73% of evaluations of false/misleading articles in the treatment group and 91% of evaluations of false/misleading articles in the control group” What’s the source of the difference here?

This is a good question, and we thank the reviewer for asking it. As discussed in the previous answer, the web extension for the treatment group was slightly different than the one for the control group. For technical reasons, respondents in the treatment group had to wait at least five seconds for the web extension to collect their Google search engine results, which may have resulted in some respondents accidentally removing the web extension. If they did not wait for five seconds on a Google search results page, the extension would turn off and they would have to manually turn it back on. These instructions were presented clearly to the respondents, but it likely resulted in this difference. We add this explanation to pages 11-12:

“This difference in compliance rates can be explained by the difference in the web extension for the treatment group relative to the one given to the control group. For technical reasons related to capturing HTML, the respondents in the treatment group had to wait at least five seconds for the web extension that was installed to collect their Google search engine results, which may have resulted in some respondents accidentally removing the web extension. If they did not wait for five seconds on a Google search results page, the extension would turn off and they would have to turn it back on. These instructions were presented clearly to the respondents, but likely resulted in differences in compliance.”

(11) “Only 12% of search queries about true articles return at least one unreliable news link among the top ten results, whereas 32% of search queries about false/misleading articles return at least one unreliable news link among the top ten results.” This reinforces the need for a comparison between true and false articles.

We completely agree with this point and believe that it is necessary to report the search effect on belief in true information. We present new analyses on the effect of searching online on belief in true news articles, which we describe in detail in response to the reviewer’s first comment.

(12) “Searching online also increased the average score by 0.15 (F=12.149, P=0.0467) on a 4-point ordinal scale and 0.16 (F=11.430, P=0.2155) on a 7-point ordinal scale” The latter isn’t an increase but is stated here as if it is.

We thank the reviewer for catching this error, which we have now changed to the following: “Searching online also increased the average score by 0.16 ($P=0.0434$, Cohen's $D = 0.16$) on a 4-point ordinal scale, but not on a 7-point ordinal scale ($P=0.2155$, Cohen's $D = 0.10$)”

Reviewer 2:

(1) Authors should characterize the stories in the text as highly popular stories, given that they were the top stories on a given day, shared from various US news sources. Obviously there are differences between low quality and high quality sources in terms of their reach, but the popularity of these stories is important to note. There's a significant difference in implications of this work if we're encountering obscure news versus widely disseminated news. Also adds the complicated reminder that popular news from US news sources routinely contains mis/disinformation.

We thank the reviewer for this comment and agree that the highly popular nature of these articles should be further emphasized. In the newly revised draft, we have replaced “popular” with “highly popular.” These revisions are made in red in the new draft.

To be clear, though, we consider this a feature of the work, not a bug – we designed our article sampling frame precisely so we would be studying news stories that were actually popular online.

(2) Authors chose to evaluate the effects of SOEM on misinformation stories. What are the effects of SOEM on true stories? Are there cases where users try to verify true stories and end up persuading themselves that they are false? While not the focus of this study, it seems important to share any insights on this topic, as one policy implication of this study is that we should be far more cautious in urging people to "do their own research". If there are significant cases where people turn away from true information (as some of the evaluation of studies 1-4 seems to suggest), that emphasizes this policy point.

Based on the response of multiple reviewers and editors, we agreed that the manuscript would be greatly strengthened by including the effect of search on evaluating true news (something we originally thought would distract from this paper’s focus on belief in false news) and have accordingly updated the manuscript to report the effect of searching online to evaluate news on belief in true news stories as well.

In the fifth section of the paper (titled “The Impact of Search on Belief in True News” on pages 19-23), we report pre-registered analyses on the effect of searching online on belief in true news in Studies 1-5. We find that while the search effect on belief in true news is similar to false/misleading news in the between respondent studies (1 and 5), the effect on true news is either smaller than on false/misleading news or non-existent across the within respondent studies

(2-4). Building off the literature on the search engine information environment, we run new exploratory analyses to investigate whether the search effect on true news has heterogeneous effects depending on the quality of the source of the new story. More specifically, we subset our measurement of the search effect on true articles from mainstream (more reputable) and low-quality (less reputable) sources. We find that the effect of online search on true news is much larger if the article is published by a low-quality source than if published by a mainstream source. In fact, the effect of searching for additional information online about a true mainstream news story is either small or non-existent. We incorporate these results into the final section of the paper. We think these heterogeneous effects paint a comprehensive and complex picture of the online search effect that, taken together, provides novel evidence with which to inform the development of digital media literacy strategies.

(3) Centering Golebiewski and boyd's notion of data voids is helpful, but it might be worth exploring their argument in more depth and considering some key implications. Golebiewski and boyd believe that right wing actors are often working to seize on data voids - they give the example of the shooting in Sutherland Springs, TX and identify situations where white supremacists created content very rapidly to pull viewers from the data void. I think what authors may be seeing is an ecosystem of mis/disinfo similar to that documented by Benkler et al in Networked Propaganda. When users are finding low quality information by searching for the title or URL of an article they're trying to evaluate, it's likely they are stepping into traps consciously set by misinformation brokers. Misinfo pages often link to one another to increase their search engine rankings and their perceived credibility. Searching for a title or URL, rather than for an article's claims, may be a particularly dangerous strategy when these misinfo ecosystems have been set up to trap the unwary.

We thank the reviewer for this comment and agree that a deeper analysis of data voids and a consideration of Benkler's work on networked propaganda add important context for our study. We added a more in-depth discussion of this literature, first on pages 10-11:

“Benkler et. al. argue that the media dynamics in the United States (particularly on the right) “tend to reinforce partisan statements, irrespective of their truth” (Page 75). This “propaganda feedback loop” creates a network of news outlets reporting the same misinformation and thus flooding search engine results with false but seemingly corroborating information.”

We also add a more detailed analysis of data voids on pages 18-19:

“Using the headline/lede as a search query likely produces unreliable results because they contain distinct phrases that only producers of unreliable information use. Golebiewski and boyd

2019 find that manipulators create content that dominates the search engine environment for people who use certain search terms. An investigation of the search terms used by individuals and the quality of news sources they are exposed to for one article in Study 5 appears to support this line of reasoning. Specifically, we analyzed the search terms for those searching online about the false/misleading article titled: “U.S. faces engineered famine as COVID lockdowns and vax mandates could lead to widespread hunger, unrest this winter.” The term “engineered famine” in the article is a unique term unlikely to be used by reliable sources. An analysis of respondents' search results finds that by adding the word “engineered” in front of “famine” changes the search results returned. 0% of search terms that contained the word “famine” without “engineered” in front of it returned unreliable results, whereas 63% of search queries that added “engineered” in front of the word “famine” were exposed to at least one unreliable result. In fact, 83% of all search terms that returned an unreliable result contained the term “engineered famine.” Over half (52%) of the respondents that used “engineered famine” in their search term were ideologically congruent to the ideological perspective of the misinformation, whereas less than a quarter of respondents (23%) who just used “famine” without “engineered” in front of it were ideologically congruent to the ideological perspective of the misinformation. There is very little difference in the average digital literacy of these groups.”

(4) How common was this search strategy? I.e., what percent of users used URLs in their searches?

We agree that this information provides a necessary frame of reference for our analysis. On page 18, we add that “Roughly 9% of all search queries that individuals entered were the exact headline or URL of the original article.” We believe that this is a significant percentage of search queries.

(5) Value of Fleiss's Kappa doesn't indicate a huge level of agreement between professional factcheckers - the authors note that this level of agreement is higher than in similar studies. Might be worth noting that it's surprisingly challenging to validate content, even by professionals, in real time. This also helps explain why it's so hard for Google and other search engines to flag misinfo, especially rapidly moving misinfo, as it's hard to get agreement that something is, in fact, misinfo.

We agree with this point and we add some discussion of this on page 31. See below:

“Although this level of agreement is quite low, it is slightly higher than other studies that have used professional fact-checkers to rate the veracity of both credible and suspect articles using the same categorical scale our fact-checkers used (Allen et. al. 2020). This low level of agreement of professionals over what is misinformation may also explain why so many respondents believe

misinformation and why searching online does not effectively reduce this problem. Identifying misinformation is a difficult task, even for professionals.”

For what it is worth, we also think this illustrates the value of relying on 4-6 fact checkers, as we do in the study. Given the fact that the classifications by the fact checkers are obviously generating some random noise in our estimates, relying on more as opposed to fewer professional fact checkers should be reducing the amount of measurement error.

(6) It seems like there's a great deal of data that didn't make it into the paper - asking users their search strategies, keywords used, etc. as well as behavioral data from study 5. Would the paper benefit from using at least some concrete examples of searches that generated higher and lower quality results? Suspect the team in planning more detailed analysis going forward - if not, releasing this data to allow more detailed analysis by collaborators would be incredibly helpful.

Yes! We plan to release anonymized data in a repository for others to use following the publication of this study. Some of it can already be viewed here:
https://github.com/SMAPPNYU/Do_Your_Own_Research

Reviewer 3:

(1) I would like to applaud the authors for performing various studies to investigate this important question. The consistency of findings observed across the studies is encouraging. However study 5 requires more work, both in terms of better articulating how the two analyses address motivated reasoning and media literacy and in terms of acknowledging alternative explanations. A bit more on this below: - If I understand the write up correctly, the authors claim that both exposure of misinfo (bad results showing up in google results) and belief in it (making the wrong assessment after seeing it in the results) can indicate motivated reasoning (in so far as there is a connection to ideology). I can see the case for the latter but I need the authors to better argue why the former is connected to motivated reasoning. Getting exposed to search results do not say anything (at least directly) about one's processing of info. Overall, a significant rewrite of this section is needed to improve its presentation. My concerns here also relate to acknowledging alternative explanations. Indeed which search results one sees for a query can be driven by personalization but a huge component of that personalization is location. The authors do not collect this data or consider it in their theoretical models. One might be served results based on their location and given the ideological geographical segregation, the factors that dictate google results might be less about the person themselves and more about where they live. There also seems to me that there is a simpler explanation of why we see worse results for people who see low credibility domains in their results. As you noted in your codebook, people are

instructed to use info even fi it is coming from a domain they don't trust. As you also noted, low credibility publishers form a network and reshare each others' content. So the closest text might be coming from those sites (even if one is not searching for the url only). Given your directions to the subjects, isn't this finding expected? This is related to media literacy dimension but I find it hard to separate this from the quality of the direction provided for the human subjects.

We thank the reviewer for this excellent comment, which we respond to in order.

First, the reviewer correctly points out that there are other mechanisms, such as personalization, that might affect exposure and magnify the impact of partisanship. We add a discussion of this possibility to the main text of the paper on page 16. Text below:

“Previous work has found that search engine results for political search queries can be personalized to individual-level characteristics and so the type of search query and the type of user may affect the results that are returned (Robertson et al., 2018), possibly amplifying partisanship (Hu et al., 2019). This may lead to a concentrated effect among those ideologically congruent to the misinformation about which they are searching.”

In addition, the reviewer’s analysis forced us to reconsider our interpretation of this concentrated exposure as the result of “motivated reasoning.” As the reviewer highlights, we cannot isolate motivated reasoning as the mechanism to explain exposure to low quality information, as our design cannot determine whether or not respondents were actually motivated to seek out ideologically aligned results. To reflect this, we replaced “motivated reasoning” with “ideological congruence.” We run analyses that demonstrate ideological congruence between the respondent’s self-reported partisanship and the ideological slant of the misinformation is associated with exposure to unreliable news sites (pages 16-17); however, as we discuss on these pages, we cannot adjudicate whether this association is the result of search strategies, motivated reasoning, or search engine personalization. To this end, we think “ideological congruence” is a more accurate description of our line of inquiry.

In regards to the instructions, we thank the reviewer for this comment and their close reading of our survey instrument. To determine whether our results were biased by the instructions we provided respondents, we ran a sixth separate study in August and September of 2022 to test three different types of instructions to search for more information online. More specifically, these instructions varied the level of detail regarding the search process. Crucially, we compared results using the original language to a version of instructions that dropped the last line about searching even for non-credible sources and another version where the instructions to search were quite short. We found similar effects for each type of instruction. We discuss this new study in a footnote on page 4 of the manuscript and provide an expanded explanation of this study and

the results in Section R of the Supplementary Materials (pages 95-99 of Supplementary Materials). In addition, we ran analyses that provide suggestive evidence of the search effect when individuals search unprompted. In most of our original studies, we asked individuals in the control group whether they searched online to evaluate information. Using these data, we found that people who searched for more information without prompting were also more likely to rate false/misleading news as true than those who did not search online. We added text discussing this analysis on page 25.

(2) -- The authors provide a clear description of the 5 studies at the end but not in the beginning. I would encourage the authors to move the description: "(1) between-respondent and within-respondent experiments (2) using general misinformation and misinformation about a salient event ..." to the intro and instead have the shorter summary currently provided in the intro to the conclusion. I found myself wondering what those 5 studies were and why they were needed. This short description would have addressed that early on.

We agree that more clarity is needed in the introduction, and we added a new paragraph detailing these studies. It can be found on pages 3-4 with the following text:

“To this end, we run five separate experiments that measure the effect of SOTEN on belief in false and true news stories. Four of these studies utilize survey experiments, while the fifth combines survey and digital trace data. In our first four studies, we measure the effect of SOTEN on belief in highly popular false and true news stories by utilizing different experimental designs (within-subjects and between subjects) and in a variety of contexts. Study 1 tests the effect of SOTEN on belief in both highly popular false/misleading and true news directly after an article’s publication (within 48 hours) using a randomized controlled trial where participants were randomly assigned to one of two groups: the treatment group, in which respondents were encouraged to search online to help them evaluate randomly assigned news articles; or the control group, in which they were not encouraged to search online. Given that consumers of false news online often encounter these stories shortly after publication, we collected respondent evaluations and digital trace data within 48 hours of publication. A study run months or years after publication would test the impact of a different information environment, and it would be impossible to replicate the original search results from the period of most likely exposure. In addition, it is important that we test the effect of SOTEN in real time because misinformation often arises in an uncertain and dynamic information environment where individuals feel a psychological need for understanding (DiFonzo and Bordia, 2007). To measure the effect of SOTEN on belief in misinformation, we run this study during the period—and thus within the information environment—that misinformation was originally generated and most likely to be

consumed. In Study 2, we test whether the effect of SOTEN can change an individual's evaluation after they had already assessed the veracity of a news story. The third study (Study 3) is another within-subjects design that measures the effect of SOTEN months after publication, rather than directly after publication, while Study 4 (also a within-subjects design) measures the effect of SOTEN on recent news about a salient topic with significant news coverage (in our case, the Covid-19 pandemic). In each study, individuals in the treatment group receive a set of recommendations, provided to us by a partner organization, encouraging the use of online search to evaluate the news articles presented in their survey instrument. In the fifth and final study, we run a between-respondent study that combines survey and web-tracking data to identify the effect of exposure to low and high quality search engine results on belief in popular misinformation within 72 hours after publication. By collecting search results using a custom web browser plug-in, we can identify how the quality of these search results may affect users' belief in the misinformation being evaluated.”

(3) -- A similar idea applies to their characterization of the misinformation assessment tasks. Given their questions and claims, it is really important to provide more information about the guidance provided to the subjects when they are making their assessments. The authors claims are tightly connected to the effectiveness of media literacy advice but this advice is rarely as simple as "do you research". It generally comes with cues to look for, definitions of credibility etc. So without information on that in the main paper, a skeptical reader might assume the authors picked a straw man version of media literacy advice and question the findings. I think there are important improvements in those codebooks that could have changed at least the magnitude of the effect sizes but this concern is not completely unaddressed by the authors. So my comment here is three-fold: 1) simply provide more information about what you have done in the intro/summary of the experiments, 2) connect that to past research and guidance and discuss how your codebooks are aligned (or not), and 3) discuss in the discussion how changes to the codebook can change some of the findings. For instance, the codebook a) doesnt provide much guidance on what sources should be trusted and b) also explicitly tells the subject to use information from sources they dont trust. These might have shaped people's assessments.

We agree that we need to provide more clarity early on, as well as identify the extent to which our instructions are (or are not) impacting our results. You can find this new description of the instructions given to the individuals on page 4 and written below:

“In each study, individuals in the treatment group receive a set of recommendations, provided to us by a partner organization, encouraging the use of online search to evaluate the news articles presented in their survey instrument.”

And as reported above, our sixth study—which tested the search effect with different instructions—provides new and necessary evidence that illustrates the generalizability of our results to different media literacy guidelines.

(4) - The authors rely on fact checkers to find a subset of articles that are deemed misinfo. But as they also state in their paper, the interrater reliability is only 0.42, much lower than what is acceptable in content analysis research. The authors then collapse the labels provided by the fact checkers to a single label. One could have instead used other measures such as increase in agreement with a random fact checker (this would mean that you don't flatten but inspect agreement with all). I would like to see the authors consider these other alternatives and discuss the limitations/implications given the disagreement observed even among the experts. There are some nice findings briefly mentioned already (how the direction of results dont change if we consider a subset of articles with stronger agreement) but expert disagreement can be considered an indicator of difficulty and the treatment effects might vary with it.

This is a great point. We add a new analysis to determine if the search effect changes as the agreement of fact-checkers that an article is false changes. To this end, we ran an interaction model and present the results in Section U of the Supplementary Materials. We find that the search effect does appear to weaken for articles that fact-checkers are most likely to agree are false/misleading. We add a discussion about this to page 31 of the main text.

“To determine if the search effect changes with the rate of agreement of fact-checkers, we ran an interaction model and present that results in Section U of the Supplementary Materials. We find that the search effect does appear to weaken for articles that fact-checkers most agree are false/misleading. Put another way, the search effect is strongest for articles in which there is less fact-checker agreement that the article is false, suggesting that online search may be especially ineffective when the veracity of articles is most difficult to ascertain.”

(5) - The authors make sweeping statements in their conclusions, in particular about media literacy campaigns. However, the guidance they provide the subjects is rather limited and various studies in content analysis would indicate that codebooks and training provided to the subjects can strongly shape their assessments and accuracy. I think that the authors can qualify their statements while not losing their power.

We agree with the reviewers assessment and we have edited the conclusion to better qualify our results. We believe these edits not only more accurately describe the implications of our study, but also provide analysis that can better inform the development of media literacy efforts. The last paragraph in our conclusion properly qualifies our results:

“The QAnon movement recommends that people ‘do the research’ themselves (Marwick and Partin, 2020), which seems like a counter-intuitive strategy for a conspiracy theory oriented movement. Our findings, however, suggest that the strategy of pushing people to verify low-quality information online might paradoxically be even more effective at misinforming them. For those who wish to learn more, they risk falling into data voids—or informational spaces where there is plenty of corroborating evidence from low-quality sources—when using online search engines, especially if they are doing ‘lazy searching’ by cutting and pasting a headline or URL. Ironically, media literacy guides also place an emphasis on doing your own research. Our findings highlight the need for media literacy programs to ground their recommendations in empirically tested interventions, as well as search engines to invest in solutions to the challenges identified here. For example, recent developments in the space — such as the expansion of teaching lateral reading strategies (Breakstone et al., 2021) and Google’s warning when no credible information is available given search queries — are interesting steps in this direction and deserve further testing.”

(6) - While the authors are primarily interested in how people assess misinformation, I think it is equally important how they assess correct information. For instance, "do your research" frames might get people to start questioning reliable information. I would like to see analysis of how the treatment affects true articles. Not all studies have this but some do.

Based on the response of multiple reviewers and editors, we agreed that the manuscript would be greatly strengthened by including the effect of search on evaluating true news (something we originally thought would distract from this paper’s focus on belief in false news) and have accordingly updated the manuscript to report the effect of searching online to evaluate news on belief in true news stories as well.

In the fifth section of the paper (titled “The Impact of Search on Belief in True News” on pages 19-23), we report pre-registered analyses on the effect of searching online on belief in true news in Studies 1-5. We find that while the search effect on belief in true news is similar to false/misleading news in the between respondent studies (1 and 5), the effect on true news is either smaller than on false/misleading news or non-existent across the within respondent studies (2-4). Building off the literature on the search engine information environment, we run new exploratory analyses to investigate whether the search effect on true news has heterogeneous effects depending on the quality of the source of the new story. More specifically, we subset our measurement of the search effect on true articles from mainstream (more reputable) and low-quality (less reputable) sources. We find that the effect of online search on true news is much larger if the article is published by a low-quality source than if published by a mainstream source. In fact, the effect of searching for additional information online about a true mainstream

news story is either small or non-existent. We incorporate these results into the final section of the paper. We think these heterogeneous effects paint a comprehensive and complex picture of the online search effect that, taken together, provides novel evidence with which to inform the development of digital media literacy strategies.

Thank you again for your constructive feedback. It was extremely helpful in improving this paper. We appreciate it. We hope that the changes to our manuscript as well as our responses to comments have sufficiently addressed the comments made. We look forward to hearing back from you.

Sincerely,

Kevin Aslett, Zeve Sanderson, William Godel, Nate Persily, Jonathan Nagler, and Joshua A. Tucker

Reviewer Reports on the First Revision:

Referees' comments:

Referee #1 (Remarks to the Author):

The authors have made a quite extensive series of revisions and, I think all can agree, the paper is really quite improved. I think the authors do provide reasonable evidence for a major (and I think important) conclusion: Search does not do well under ambiguity. Critically, and unfortunately, search is also the most desirable and (probably) used when ambiguity is high.

The authors have definitely done more to highlight that their research is really about searching given ambiguity (e.g., more explanation that search is happening in a relatively short time period after an article is published). However, I continue to think that they need to go a bit further. As I noted in my past review, this is essentially an existence proof: The authors set up a (reasonable) set of scenarios where search occurs. The problem is that it's very hard to know how representative these scenarios are of actual ("real life") search behavior, and therefore I think the paper should be framed around how search *can* increase belief in misinformation (and not that it *does*).

Just to take a concrete example, from the abstract: "Across five experiments, we present consistent evidence that online search to evaluate the truthfulness of false news articles actually increases the probability of believing misinformation." Instead of "actually increases" I think it should read "can actually increase". Subtly but very important difference, in my view.

Related to this, and in response to my comment about making it more clear in the manuscript that the study is necessarily historical, the authors have added a footnote. I don't think this goes far enough. Instead, I think they should essentially repeat throughout the manuscript a qualification that the data apply to "for the point-in-time investigated".

These revisions would be quite minor but I think they are necessary.

Also, the effect sizes are very small. I think they should be stated in abstract and this point should be acknowledged explicitly throughout.

Referee #3 (Remarks to the Author):

I would like to thank the authors for the significant amount of work they put into this revision. Almost all of my comments are fully addressed, and the rest are at least partially addressed. I think the paper is almost ready to publish, but I will highlight two of my comments that were partially addressed. Both of those can be addressed through writing (no new analyses are needed).

- I still have an issue with a simplistic characterization of existing media literacy efforts. I really appreciate that the authors carried out a 6th study to vary the instructions given. But the instructions are still not really aligned with the rather detailed instructions I see across multiple media literacy efforts. In such efforts, the reader is generally instructed to trace claims, check their emotions, check the credibility of sites, etc in media literacy education. I think the last sentence added to the conclusion is great, but it makes it sound like most other media literacy efforts just instruct people to search and only search. I don't think this is a fair characterization of media literacy related to misinformation. I would strongly encourage the authors to tone down their claims around media literacy throughout the paper. You can clarify the fact that here you are testing the effect of searching with limited guidance. Searching with the added guidance could lead

to different results (or perhaps not if people are generally ignoring the added instructions but that is not tested here).

- Minor: The authors mention individual-level personalization in the section titled "Why Are Individuals Exposed to Unreliable Information In Their Search Results?" Individual-level personalization is indeed likely to lead to ideologically congruent results being returned as search results. But as I mentioned before, location-based personalization is the most common personalization technique observed. I would acknowledge that (eg., Hannak, A., Sapiezynski, P., Molavi Kakhki, A., Krishnamurthy, B., Lazer, D., Mislove, A. and Wilson, C., 2013, May. Measuring personalization of web search. In Proceedings of the 22nd international conference on World Wide Web (pp. 527-538)). This is because individual vs location personalization has different implications for what search engines are doing wrong here (even if google stops using individual level personalization, such ideologically congruent results are likely to still show up in results due to location based personalization).

A few other things that would be nice to have, if possible:

- Given the results for true news, 1-2 sentences about implications in discussion might be warranted: It seems like search is bad in the sense that it can increase trust in false information, but it is also good in the sense that it increases trust in true news. Any thoughts on what the net effect is? This depends on how people generally use search (more often to look up false or true news) and what our priorities are in relation to trust in news and institutions.

- Especially given the results for true news, I find myself wondering how the SOTEN effects vary across parties (or ideologies). This is particularly important given that misinfo effects are asymmetric. I think the paper has made substantial contributions already, so I wouldn't say such an analysis is required to publish the paper,

Thanks again for all the effort. I really liked reading this paper and believe it will be a great contribution to the field.

Author Rebuttals to First Revision:

To the Editors of *Nature*,

Please find attached the revised manuscript “Do Your Own Research: How Searching Online to Evaluate Misinformation Can Increase Its Perceived Veracity” for consideration for publication in *Nature*.

We thank the editor for the opportunity to revise our paper along the lines suggested. We addressed each individual reviewer comment. We detail these changes below. We thank the editor and reviewers for their generous and detailed feedback throughout this process — we hope you and they will share our assessment that our responses to their suggestions added clarity and depth to our manuscript.

Reviewer 1:

- (1) The authors have definitely done more to highlight that their research is really about searching given ambiguity (e.g., more explanation that search is happening in a relatively short time period after an article is published). However, I continue to think that they need to go a bit further. As I noted in my past review, this is essentially an existence proof: The authors set up a (reasonable) set of scenarios where search occurs. The problem is that it’s very hard to know how representative these scenarios are of actual (“real life”) search behavior, and therefore I think the paper should be framed around how search **can** increase belief in misinformation (and not that it **does**).**

Just to take a concrete example, from the abstract: “Across five experiments, we present consistent evidence that online search to evaluate the truthfulness of false news articles actually increases the probability of believing misinformation.” Instead of “actually increases” I think it should read “can actually increase”. Subtly but very important difference, in my view.

We agree and have edited the title of the manuscript to “Online Searches to Evaluate Misinformation Can Increase Its Perceived Veracity.” We made the specific change reviewer 1 alluded to and we also changed this throughout the draft. For example, we edited statements such as “searching makes study participants more likely to believe that true news stories are true” to “searching can make study participants more likely to believe that true news stories are true”. We will note, though, that when directly describing the results of the findings – as opposed to the implications of the studies – we continue to use more definitive language, as the more qualified language (“can increase”) does not make sense in the context of discussing empirical findings.

- (2) Related to this, and in response to my comment about making it more clear in the manuscript that the study is necessarily historical, the authors have added a footnote. I don’t think this goes far enough. Instead, I think they should essentially**

repeat throughout the manuscript a qualification that the data apply to “for the point-in-time investigated”.

In four different places we have made this revision. For example we now write that “the effect of SOTEN on belief in false and true news stories for the point-in-time investigated.” We agree that these effects could very well change in the future.

(3) Also, the effect sizes are very small. I think they should be stated in abstract and this point should be acknowledged explicitly throughout.

We added the effect size in percentage to the abstract and make the effect sizes explicitly clear throughout the paper.

Reviewer 3:

(1) I still have an issue with a simplistic characterization of existing media literacy efforts. I really appreciate that the authors carried out a 6th study to vary the instructions given. But the instructions are still not really aligned with the rather detailed instructions I see across multiple media literacy efforts. In such efforts, the reader is generally instructed to trace claims, check their emotions, check the credibility of sites, etc in media literacy education. I think the last sentence added to the conclusion is great, but it makes it sound like most other media literacy efforts just instruct people to search and only search. I don't think this is a fair characterization of media literacy related to misinformation. I would strongly encourage the authors to tone down their claims around media literacy throughout the paper. You can clarify the fact that here you are testing the effect of searching with limited guidance. Searching with the added guidance could lead to different results (or perhaps not if people are generally ignoring the added instructions but that is not tested here).

We agree that this is an important addition. In the discussion we added the following text to make our recommendation more clear:

“It should also be noted that a number of media literacy education programs provide a larger set of instructions in addition to the search recommendation; however, given the prevalence of the search recommendation across media literacy interventions and the ease with which people can adopt the recommendation, we think it is important to understand the effect of online search with limited guidance.”

- Minor: The authors mention individual-level personalization in the section titled "Why Are Individuals Exposed to Unreliable Information In Their Search Results?" Individual-level personalization is indeed likely to lead to ideologically congruent results being returned as search results. But as I mentioned before, location-based personalization is the most common personalization technique observed. I would acknowledge that (eg.,

Hannak, A., Sapiezynski, P., Molavi Kakhki, A., Krishnamurthy, B., Lazer, D., Mislove, A. and Wilson, C., 2013, May. Measuring personalization of web search. In Proceedings of the 22nd international conference on World Wide Web (pp. 527-538)). This is because individual vs location personalization has different implications for what search engines are doing wrong here (even if google stops using individual level personalization, such ideologically congruent results are likely to still show up in results due to location based personalization).

We agree that this is important to add. Therefore the sentence referenced has been revised to the following:

“Relatedly, although research shows that the most common form of personalization is location-based personalization (Kliman-Silver et al., 2015), search engine results for political search queries can be personalized to individual-level characteristics and so the user's ideology may lead to more information that aligns with their ideological worldview (Robertson et al., 2018), possibly amplifying the impact of ideological congruence (Hu et al., 2019).”

A few other things that would be nice to have, if possible:

- Given the results for true news, 1-2 sentences about implications in discussion might be warranted: It seems like search is bad in the sense that it can increase trust in false information, but it is also good in the sense that it increases trust in true news. Any thoughts on what the net effect is? This depends on how people generally use search (more often to look up false or true news) and what our priorities are in relation to trust in news and institutions.

We agree that this should be highlighted. The end of the second paragraph in the discussion section has been revised to the following:

“Finally, we find evidence that SOTEN increases belief in true news from low-quality sources, but inconsistent evidence of the effect of SOTEN on belief in true news from mainstream sources. The implications of these heterogeneous effects across article veracity and source quality will depend on how people use search engines (i.e. the prevalence of searching about false or true news) and our priorities in relation to building trust in news and institutions. While practitioners and policymakers must balance the heterogeneous effects of SOTEN across article veracity and source quality, we think the increase of belief in misinformation should be of particular import when designing digital media literacy interventions that recommend search as a potential strategy.”

- Especially given the results for true news, I find myself wondering how the SOTEN effects vary across parties (or ideologies). This is particularly important given that misinfo effects are asymmetric. I think the paper has made substantial contributions already, so I wouldn't say such an analysis is required to publish the paper.

This is a very interesting point about asymmetric effects. We ran some new analyses looking at the effect across different political ideologies and didn't find a significant difference. We run two different analyses shown in extended data figure 5 and extended figure 10 that display this. These are referenced in the paper on pages 12 and 13.

Page 12: "Across all five studies, we do not find a statistically significant differential effect of SOTEN by political ideology or ideological congruence to the news article (see Extended Data Fig. 5)."

Page 13: "Interestingly, We do not find a statistically significant differential effect of low-quality information on belief across different levels of ideological congruence to the news article (see Extended Data Fig. 10)."